# ScDiVa: Masked Discrete Diffusion for
# Joint Modeling of Single-Cell Identity and Expression

**Mingxuan Wang** [1 2 3 4]  **Gaoyang Jiang** [5]  **Zijia Ren** [6]  **Cheng Chen** [7]  **Chuangxin Zhao** [8]  **Lu Shi** [7]  **Yanbiao Ma** [1 3 4 *]

## Abstract

Single-cell RNA-seq profiles are high-dimensional, sparse, and unordered, causing autoregressive generation to impose an artificial ordering bias and suffer from error accumulation. To address this, we propose scDiVa, a masked discrete diffusion foundation model that aligns generation with the dropout-like corruption process by defining a continuous-time forward masking mechanism in token space. ScDiVa features a bidirectional denoiser that jointly models discrete gene identities and continuous values, utilizing entropy-normalized serialization and a latent anchor token to maximize information efficiency and preserve global cell identity. The model is trained with depth-robust time sampling and a dual denoising objective, exposing it to varying sparse-observation regimes while encouraging accurate recovery of both identity and magnitude. Pre-trained on 59 million cells, scDiVa achieves strong transfer performance across major benchmarks, including batch integration, cell type annotation, and perturbation response prediction. These results suggest that masked discrete diffusion serves as a biologically coherent and effective alternative to autoregression.

---

[*]Corresponding author. [1]Gaoling School of Artificial Intelligence, Renmin University of China, Beijing, China [2]School of Statistics, Renmin University of China, Beijing, China [3]Beijing Key Laboratory of Research on Large Models and Intelligent Governance [4]Engineering Research Center of Next-Generation Intelligent Search and Recommendation, MOE [5]School of Computer Science and Technology, Huazhong University of Science and Technology [6]School of Mathematics, Jilin University [7]CRE Life Institute [8]Beijing Academy of Artificial Intelligence. Correspondence to: Yanbiao Ma <ybma1998@ruc.edu.cn>.

*Proceedings of the $43^{rd}$ International Conference on Machine Learning*, Seoul, South Korea. PMLR 306, 2026. Copyright 2026 by the author(s).

## 1. Introduction

The advancement of single-cell RNA sequencing (scRNA-seq) has necessitated scalable "Foundation Models" capable of extracting cellular representations from global datasets. Inspired by Large Language Models (LLMs), Transformer-based methods such as scBERT (Yang et al., 2022), Geneformer (Theodoris et al., 2023), and scGPT (Cui et al., 2024) treat genes as discrete "tokens", showing potential in cross-task transfer through self-supervised learning.

However, applying NLP paradigms to genomics faces a fundamental structural mismatch. First, gene expression profiles are high-dimensional, unordered multisets; Autoregressive (AR) models enforce an artificial sequential order that disrupts symmetric gene regulatory interactions (Cui et al., 2024). Second, existing tokenization strategies struggle to balance sequence length with sparsity, often capturing binary gene presence but failing to reconstruct precise continuous expression intensities (numerical fidelity). Meanwhile, continuous generative models like scVI (Lopez et al., 2018) (VAEs) tend to produce over-smoothed results, and existing diffusion models (Luo et al., 2024) relying on Gaussian noise often fail to model discrete events and stochastic dropout of single-cell data.

To address these "sequence-set" and "discrete-continuous" challenges, we propose Single-cell Masked Diffusion for Identity & Value (**scDiVa**), a foundation model using a Masked Discrete Diffusion framework (Austin et al., 2021). Unlike Gaussian-based methods, we formalize a correspondence between absorbing-state corruption and dropout-like gene non-detection. By using an absorbing `[MASK]` state to simulate signal loss, scDiVa jointly learns gene-gene association patterns and quantitative expression manifolds in one probabilistic framework.

ScDiVa incorporates Entropy-Normalized Serialization to allocate a fixed token budget toward informative genes under sparse long-tail observations. Furthermore, we design a Dual Denoising Loss that constrains topological classification and dosage regression. This architecture leverages bidirectional non-causal context, supporting accurate recovery of cellular states in both "Rank" and "Value" dimensions while reducing artificial sequential bias.

Our contributions are threefold:

- We propose scDiVa, a foundation model grounded in masked discrete diffusion. By aligning absorbing-state corruption with dropout-like missingness, scDiVa reduces the ordering bias introduced by autoregressive modeling and provides a biologically motivated alternative to Gaussian perturbation in token space.

- We employ Entropy-Normalized Serialization as an information-allocation strategy under a fixed token budget. A Dual Denoising Loss simultaneously optimizes gene identity and dosage, enabling high-fidelity recovery in Rank and Value dimensions.

- We introduce a depth-robust time-sampling strategy that exposes the model to a continuum of sparse-observation regimes. Rather than serving as an exact simulator of library-size variation, this strategy improves robustness to dropout-like missingness and supports stable transfer across datasets with heterogeneous sequencing depth.

**Conflict of Interest Disclosure.** The authors declare no financial conflicts of interest related to this work.

## 2. Related Work

### 2.1. LLM-Based Single-Cell Pre-training

Inspired by pre-training in Natural Language Processing (NLP), single-cell foundation models typically represent each cell as a sequence or a finite set of gene tokens. These models employ self-supervised objectives, such as masked prediction, to learn contextual dependencies and universal embeddings. For instance, scBERT is pre-trained on large-scale unlabeled scRNA-seq data, demonstrating advantages in few-shot fine-tuning and offering interpretability cues via attention weights (Yang et al., 2022). Similarly, Geneformer emphasizes learning contextual dependencies across massive corpora and leverages transferability for diverse network biology predictions (Theodoris & Ellinor, 2023; Theodoris et al., 2023). Building on this, scGPT extends generative pre-training to large-scale single-cell and even multi-omics data, systematically validating its transfer efficacy in tasks such as integration and annotation (Cui et al., 2024). Meanwhile, scFoundation enhances universal cell and gene representations by utilizing a larger corpus and a more comprehensive gene space, followed by evaluation across multiple tasks (Hao et al., 2024). A critical tension inherent in this approach lies in the fact that tokenization necessitates a complex trade-off among finite sequence length, extreme sparsity, and long-tail dynamic ranges. Consequently, these methods often excel at capturing the identity structure concerning which gene events occur, yet they require additional mechanisms to compensate for the numerical fidelity of continuous expression intensities.

### 2.2. Continuous Probabilistic Representation Learning

Another primary stream focuses on probabilistic generative modeling centered on continuous count distributions. A representative example is scVI, which unifies sequencing noise, batch effects, and latent space inference via Variational Autoencoders (VAEs), serving as a foundational framework for denoising and integration (Lopez et al., 2018). In multi-omics scenarios, totalVI further incorporates modalities such as RNA and proteins into a unified probabilistic model, demonstrating the transferability of combining continuous distribution modeling with a unified latent space (Gayoso et al., 2021). Concurrently, certain large-scale pre-training methods learn representations directly in the continuous domain for multi-task transfer. For instance, CellFM reports competitive results across various tasks, including perturbation-related ones, while emphasizing a scalable and efficient backbone and training scheme (Zeng et al., 2025). However, a common risk associated with continuous approaches is that if the objective function primarily encourages numerical reconstruction or regression, the model tends to heavily bias towards smooth or averaged interpretations. This results in a lack of explicit generative pressure regarding which specific gene events should be recovered when missing, thereby limiting capabilities in controllable completion and structured generation.

### 2.3. Diffusion Models and Masked Generation

Diffusion models offer a unified perspective for both continuous and discrete generation and have begun to be applied to single-cell generation and imputation. For example, scDiffusion utilizes conditional diffusion to generate high-fidelity expression profiles (Luo et al., 2024), while scVAEDer explores combining diffusion with VAEs to enhance generation and reconstruction stability (Sadria & Layton, 2025). However, most single-cell diffusion models are still based on continuous noise assumptions, often requiring additional designs to handle the discrete structure of event occurrence or disappearance. In contrast, discrete diffusion (D3PM) and masked generation (MaskGIT) define masking and reverse denoising directly within the discrete state space. These methods interpret the recovery of masked positions as a single-step prediction in the reverse process (Austin et al., 2021; Chang et al., 2022). Recent efforts to simplify and unify discrete masked diffusion objectives have further enhanced training stability and conceptual consistency (Shi et al., 2024). In the language domain, LLaDA has further validated that diffusion-based pre-training, characterized by a random masking forward process combined with Transformer-based reverse prediction, can serve as an alternative paradigm to autoregression. This approach emphasizes closed-loop consistency from the training objective to generative inference (Nie et al., 2025).

# 3. ScDiVa: Diffusion Foundation Model for Single Cells

ScDiVa is a generative foundation model designed to address the structural mismatch between autoregressive sequence modeling (Floridi & Chiriatti, 2020) and the unordered, sparse nature of single-cell transcriptomic data (Cui et al., 2024). Rather than imposing an artificial gene ordering, scDiVa models generation as a bidirectional denoising process over masked gene tokens. By formulating generation through a masked discrete diffusion process with an absorbing state (Austin et al., 2021; Shi et al., 2024), the model naturally aligns with stochastic signal dropout observed in single-cell sequencing. This framework enables joint modeling of gene identity and expression magnitude within a unified probabilistic formulation. A detailed biological interpretation of this formulation is provided in Appendix A.1.

## 3.1. Problem Formulation and Diffusion Modeling

Traditional continuous diffusion models typically operate in Euclidean space by adding Gaussian noise (Ho et al., 2020). This introduces an inductive bias of ordinality, where distances between values are assumed to be semantically meaningful. Such an assumption is incompatible with the categorical nature of discrete gene tokens. To address this mismatch, we adopt a state-transition-based masked discrete diffusion paradigm (Austin et al., 2021; Shi et al., 2024).

Formally, let $\mathbf{x}_0 = [x_1, \ldots, x_L]$ denote the discrete gene sequence representing a cell state, where each gene token $x_i$ belongs to a finite vocabulary $\mathcal{V}$. We define the forward diffusion as a continuous-time Markov process $t \in [0, 1]$ that progressively destroys information. Unlike the local noise injection in continuous models, we employ a global stochastic corruption mechanism based on an absorbing state. At any arbitrary time $t$, a token $x_t^i$ either retains its original state $x_0^i$ or transitions to the absorbing state $[\text{MASK}]$ (denoted as $\varnothing$) with probability defined by:

$$q(x_t^i|x_0^i) = (1 - t) \cdot \delta(x_t^i, x_0^i) + t \cdot \delta(x_t^i, \varnothing). \quad (1)$$

This process creates a trajectory from a fully determined profile at $t = 0$ to a state of maximum entropy at $t = 1$. This formulation closely mirrors the "dropout" phenomenon in single-cell sequencing (Hicks et al., 2018), where valid signals are stochastically lost, thereby grounding our mathematical noise model in a dropout-like missingness abstraction observed in single-cell data (see Appendix A.2 and A.8 for the formal correspondence).

The generative capability of scDiVa is realized through the reverse denoising process, which learns the conditional distribution $p_\theta(\mathbf{x}_0 \mid \mathbf{x}_t)$. By reconstructing original gene states from masked inputs, scDiVa performs bidirectional, non-causal modeling (Devlin et al., 2019), enabling each gene to be inferred from global context. This design avoids the artificial ordering and asymmetric dependencies imposed by autoregressive factorizations (e.g., $\prod p(x_i \mid x_{<i})$), which are biologically implausible given the non-sequential and symmetric nature of gene regulatory interactions (Levine & Davidson, 2005). As a result, scDiVa directly models the joint distribution $p(\mathbf{x})$ in a manner aligned with the unordered, multiset structure of gene expression profiles (Hao et al., 2024). A formal comparison with autoregressive and Gaussian diffusion models is provided in Appendix A.3.

## 3.2. Cell Representation and Unified Embedding

The adaptation of transcriptome data to Transformer architectures is non-trivial due to the inherent sparsity and long-tail distribution of gene expression. To maximize effective supervision within a finite context window $L$, we propose a specialized representation protocol that prioritizes information density over raw throughput.

Standard "ranking by expression" approaches are suboptimal because they are often dominated by high-expression but low-entropy "housekeeping" genes, which contribute minimal discriminative information regarding cell identity (Tang & Han, 2026). To counter this, we implement *Entropy-Normalized Serialization*. We define a ranking score $r_k$ that penalizes ubiquitous expression features using population-level Shannon entropy $H(g)$ (Brennecke et al., 2013):

$$r_k = \frac{v_k}{H(g_k) + \epsilon}. \quad (2)$$

This formulation introduces a data-centric inductive bias, effectively filtering out biological background noise. It compels the model to allocate its computational budget to genes with the highest discriminative power (Stuart et al., 2019), ensuring that the limited token space encodes the maximum possible cell-type-specific information (see Appendix A.4 for detailed motivation).

Following feature selection, we construct a Unified Embedding that respects the unordered nature of the data. Since absolute positions in a gene sequence are biologically meaningless, our input embedding $\mathbf{h}^{(0)}$ integrates only gene identity and expression magnitude:

$$\mathbf{h}^{(0)} = \text{Emb}gene(g) + \text{MLP}val(v). \quad (3)$$

While we discard absolute positional encodings to avoid imposing an artificial rigid Cartesian structure, capturing the intrinsic relationships between genes remains crucial. We therefore strategically inject relative position information via Rotary Positional Embedding (RoPE) in the attention layers. This allows scDiVa to robustly learn hierarchical gene-gene dependencies without overfitting to a spurious permuted sequence order (further discussion on Transformer inductive bias is provided in Appendix A.5).

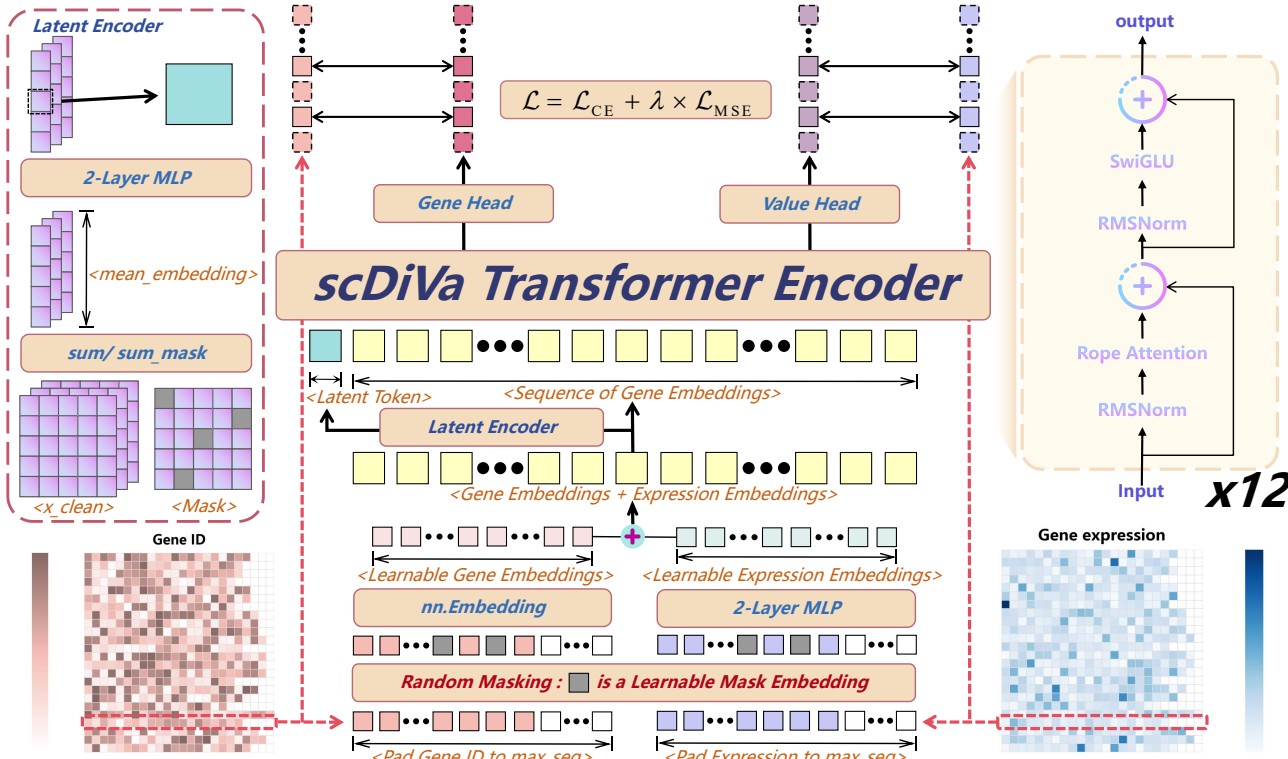

*Figure 1.* **Overview of the scDiVa Architecture.** The framework employs a masked modeling approach with a Latent Encoder to capture global cell contexts. The input gene expression profile is randomly masked and processed through a 12-layer Transformer encoder equipped with RoPE attention and SwiGLU activation. The model optimizes a dual objective ($\mathcal{L}$), combining Cross-Entropy ($\mathcal{L}_{CE}$) for gene identity reconstruction and Mean Squared Error ($\mathcal{L}_{MSE}$) for expression value regression.

To further stabilize the learning process, particularly in the high-noise regimes ($t \to 1$) where the input signal is sparse, we introduce a *Latent Variable Anchor Token* ([LAT]). In the encoder, [LAT] aggregates global information via self-attention to form a state vector $\mathbf{z}_{lat}$ (Wang et al., 2021). During the reverse denoising process, $\mathbf{z}_{lat}$ functions as a "Global Prompt", anchoring the generation to the underlying cell-state manifold. This prevents "posterior collapse" often observed in conditional generation (Higgins et al., 2017), enabling scDiVa to maintain identity coherence even when 90% of the gene tokens are masked (see Appendix A.6).

This architecture offers distinct theoretical advantages over traditional AR paradigms. We do not claim strict permutation invariance, since scDiVa still uses deterministic serialization and RoPE. Instead, the model reduces ordering bias by avoiding causal masking and by allowing every gene token to condition on bidirectional context. The parallel denoising mechanism mitigates exposure bias (Bengio et al., 2015), reducing the risk that local prediction errors propagate sequentially. Finally, the model supports omnidirectional information flow, allowing gene-gene dependencies and feedback-like regulatory patterns to be represented without imposing an autoregressive order.

### 3.3. Depth-Robust Sampling and Optimization

We conceptualize training as a denoising procedure for sparse single-cell observations, rather than as a simulator of the physical sequencing process. A reduction in sequencing depth globally thins molecular counts and is affected by library size, capture efficiency, amplification bias, and biological variation. Our masked corruption does not model all of these mechanisms. Instead, it approximates one observational consequence of shallow sequencing, namely increased gene non-detection or dropout-like missingness.

To expose the model to a continuum of sparse-observation regimes, we sample the diffusion time step $t \sim \mathcal{U}(0, 1)$ continuously (Dieleman et al., 2022), rather than using a fixed masking rate. Larger values of $t$ correspond to higher gene-token masking probabilities, whereas smaller values preserve observed signals. This training strategy encourages representations that are stable across varying observation sparsity levels and heterogeneous sequencing-depth conditions (Gayoso et al., 2022; Wu et al., 2025a). We provide additional discussion in Appendix A.7 and a mixed global-scaling-plus-masking ablation in Table 13 of Appendix G.4.

The optimization objective is governed by a Dual Denoising Loss, which imposes simultaneous constraints on the

masked set $\mathcal{M}$ to recover both gene identity and expression magnitude. The loss function $\mathcal{L}$ is formulated as:

$$\mathcal{L} = \mathbb{E}_{t, \mathbf{x}_0} \left[ \sum_{i \in \mathcal{M}} \underbrace{-\log p_\theta(g_i \mid \mathbf{x}_t)}_{\mathcal{L}_{\mathrm{id}}} + \lambda \underbrace{\|\hat{v}_i - v_i\|^2}_{\mathcal{L}_{\mathrm{val}}} \right] . \quad (4)$$

Here, the classification term $\mathcal{L}_{\mathrm{id}}$ encourages recovery of the masked gene identities, while the regression term $\mathcal{L}_{\mathrm{val}}$ enforces accurate inference of expression dosages (Eraslan et al., 2019). We provide a derivation showing that this objective corresponds to an estimator of the reverse denoising term in a variational lower bound in Appendix A.9. Minimizing this joint objective encourages scDiVa to learn cell representations that are robust to dropout-like missingness while preserving quantitative expression information.

### 3.4. Model Architecture and Pre-training Protocol

ScDiVa is instantiated as a 12-layer bidirectional Transformer encoder with approximately 94.5M parameters and a comprehensive vocabulary of 41,818 gene tokens. The complete workflow is shown in Figure 1. Feed-forward layers employ SwiGLU activations, and Pre-RMSNorm (Zhang & Sennrich, 2019) is applied throughout for enhanced numerical stability, with all accumulation performed in float32 precision (detailed architecture configurations are listed in Tables 5 and 6 in Appendix C.4). The complete training and inference algorithms are provided in Algorithm 1 and 2 (Appendix A.10).

The model is pre-trained on a corpus of 59,162,450 single-cell transcriptomes, sourced from a large proprietary dataset spanning diverse tissues. Following standard filtering and log-normalization, we apply entropy-normalized serialization to retain the top 1,200 genes. Training is conducted on four NVIDIA A100-SXM4-40GB GPUs using a global batch size of 768 via gradient accumulation (Rasley et al., 2020) and proceeds for four epochs under the depth-robust sparse-observation sampling regime. Additional details on dataset composition, entropy-normalization math, and pre-processing are reported in Appendix B.

## 4. Experiments

In this section, we present an evaluation of scDiVa across four tasks of increasing complexity. We begin with rank-value reconstruction (Eraslan et al., 2019) to validate intrinsic co-expression patterns, followed by multi-batch integration (Gayoso et al., 2022) to assess robustness against technical noise. We then evaluate cell type annotation (both fine-tuning and zero-shot) to test discriminative power, and finally gene perturbation prediction (Lotfollahi et al., 2023) to evaluate perturbation response modeling. Extensive benchmarking against state-of-the-art foundation models (Hao et al., 2024) confirms scDiVa's ability to bridge high-fidelity

*Table 1.* **Rank-Value Joint Reconstruction across datasets.** Lower L-Dist is better; higher BLEU and Spearman are better.

| Dataset | Model | L-Dist ↓ | BLEU ↑ | Spearman ↑ |
|---------|-------|----------|--------|------------|
| PBMC12k | GeneMamba_U | 430 | 0.532 | 0.469 |
| | Geneformer | 23 | 0.968 | 0.703 |
| | GeneMamba | 6 | **0.987** | 0.711 |
| | scDiVa | **5** | **0.987** | **0.812** |
| Pancreas | GeneMamba_U | 370 | 0.524 | 0.461 |
| | Geneformer | 25 | 0.956 | 0.763 |
| | GeneMamba | **12** | **0.991** | 0.792 |
| | scDiVa | 13 | 0.965 | **0.812** |
| Zheng68k | GeneMamba_U | 432 | 0.581 | 0.503 |
| | Geneformer | 25 | 0.937 | 0.901 |
| | GeneMamba | 11 | **0.996** | 0.980 |
| | scDiVa | **9** | 0.992 | **0.994** |
| Immune | GeneMamba_U | 468 | 0.659 | 0.442 |
| | Geneformer | 17 | 0.962 | 0.823 |
| | GeneMamba | 12 | **0.998** | 0.844 |
| | scDiVa | **4** | 0.997 | **0.970** |

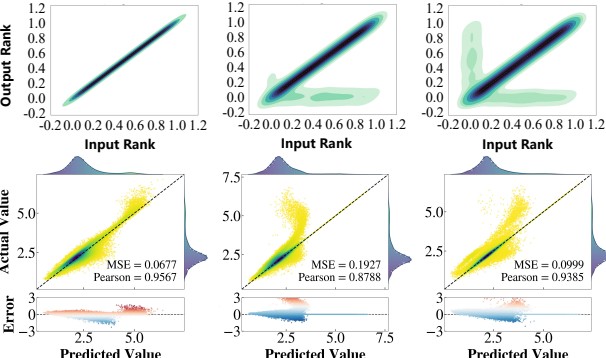

*Figure 2.* **Visual comparison of generative dynamics.** From left to right: ground-truth profile, one-step denoising, and 32-step diffusion reconstruction.

generation with precise biological discrimination.

### 4.1. Downstream Tasks and Comparative Methods

To systematically evaluate scDiVa, we established a benchmark spanning reconstruction, integration, annotation, perturbation prediction, and regulatory hypothesis generation. We used PBMC12k and Immune for *gene reconstruction*. For *multi-batch integration*, we evaluated COVID-19, Immune, PBMC12k, Perirhinal Cortex, and BMMC. For *cell type annotation*, we considered fine-tuning on hPancreas, MS, Myeloid, and Myeloid_b, together with zero-shot evaluations on Cell_Lines, DC, MCA, PBMC368K, HumanPBMC, and Pancrm. We further used Adamson and Norman for *gene perturbation prediction*, and used attention-derived gene association analysis for *regulatory hypothesis generation*.

Unless otherwise stated, the cell representation is taken from the final hidden state of the latent anchor token [LAT]. Rank-value reconstruction directly uses the pre-trained iden-

*Table 2.* **Multi-batch integration benchmark.** Avg-batch measures batch mixing, and Avg-bio measures biological conservation. We report the original comparison set together with additional recent foundation-model baselines evaluated under matched preprocessing, splits, metrics, and model-selection settings.

| Model | Immune | | PBMC12k | | BMMC | | Perirhinal Cortex | | COVID-19 | |
|---|---|---|---|---|---|---|---|---|---|---|
| | Batch | Bio | Batch | Bio | Batch | Bio | Batch | Bio | Batch | Bio |
| Harmony | 0.9514 | 0.6945 | 0.9341 | 0.7990 | 0.8999 | 0.6316 | 0.9442 | 0.8595 | 0.8781 | 0.4468 |
| Geneformer | 0.8153 | 0.6983 | 0.9545 | 0.7891 | 0.7720 | 0.6324 | 0.9127 | 0.8547 | 0.8240 | 0.5567 |
| scGPT | 0.9194 | 0.7879 | 0.9755 | 0.9018 | 0.8431 | 0.6576 | 0.9600 | 0.9552 | 0.8625 | 0.6476 |
| scFoundation | 0.8904 | 0.7337 | 0.9628 | 0.8662 | 0.7598 | 0.5250 | 0.9560 | 0.9606 | 0.8346 | 0.5468 |
| GeneMamba | 0.9536 | **0.8131** | 0.9604 | 0.8344 | 0.9157 | 0.7628 | **0.9673** | 0.9062 | 0.8742 | 0.5537 |
| CellFM | 0.9523 | 0.7934 | 0.9862 | **0.9741** | 0.9562 | 0.8014 | 0.9621 | 0.9692 | 0.9143 | 0.6415 |
| UCE | 0.9397 | 0.7482 | 0.9764 | 0.9323 | 0.9011 | 0.7225 | 0.9496 | 0.9281 | 0.8924 | 0.5916 |
| scELMO | 0.9342 | 0.7355 | 0.9726 | 0.9214 | 0.8893 | 0.7012 | 0.9445 | 0.9211 | 0.8842 | 0.5764 |
| GeneCompass | 0.9421 | 0.7684 | 0.9782 | 0.8963 | 0.9124 | 0.7581 | 0.9553 | 0.9442 | 0.9012 | 0.6093 |
| **scDiVa** | **0.9555** | 0.7785 | **0.9960** | 0.9566 | **0.9734** | **0.8712** | 0.9542 | **0.9895** | **0.9538** | **0.6689** |

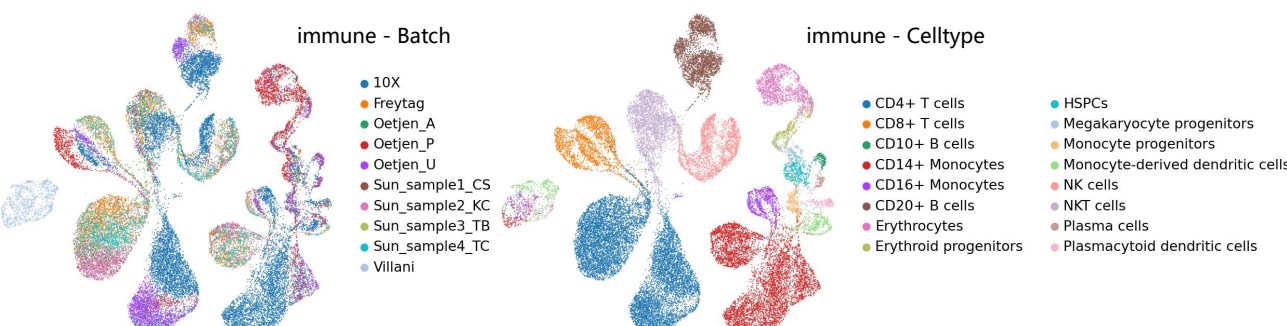

*Figure 3.* **Visualization of multi-batch integration on the Immune dataset.** The UMAP projections illustrate the latent representations learned by scDiVa. The left panel is colored by batch source, demonstrating the effective removal of batch effects (mixing). The right panel is colored by cell type, highlighting the distinct preservation of biological clusters and cellular identities.

tity and value denoising heads. Multi-batch integration fine-tunes the full encoder with reconstruction, batch-adversarial, and supervised contrastive objectives when reference cell-type labels are available; the full objective is defined in Appendix D.1. Cell type annotation freezes the first eight Transformer layers and fine-tunes the last four layers with an MLP classifier. Perturbation prediction injects perturbation identity through a dedicated [PERT] token and fine-tunes the encoder with a regression head under perturbation-level splits. The corresponding optimization hyperparameters, updated modules, task heads, losses, and model-selection criteria are summarized in Appendix Table 7. We benchmarked against representative foundation models and task-specific methods under matched protocols, including Geneformer (Theodoris et al., 2023), Harmony, CellFM (Zeng et al., 2025), scBERT (Yang et al., 2022), GeneCompass (Yang et al., 2024), UCE (Rosen et al., 2023), scELMO (Liu et al., 2026), SVM, scmap (Abdelaal et al., 2019), and GEARS (Roohani et al., 2024). Detailed dataset statistics, metric definitions, and extended task protocols are provided in Table 4 (Appendix B.4) and Appendix E.

## 4.2. Rank-Value Joint Reconstruction

Gene reconstruction requires restoring both relative Rank (robust against scaling noise, capturing underlying topology) and absolute Value (enforcing magnitude constraints). This dual focus effectively prevents "pseudo-consistency" where correct ranking masks intensity collapse.

We evaluate generative capability using one-step denoising, where the model predicts gene-value pairs from partially observed inputs and ranks reconstructed genes by model probability. Performance is comprehensively assessed using L-Dist (distribution shifts), BLEU (sequence matching), and Spearman coefficient (rank correlation).

Table 1 highlights model performance across four datasets. GeneMamba_U performs poorly, lacking fine-grained constraints. While Geneformer and GeneMamba achieve high BLEU scores (0.94–0.998), they fail to fully capture biological rank structures. In contrast, scDiVa excels in Rank while maintaining high BLEU, achieving record Spearman correlations on Immune **(0.970, +14.9%)** and PBMC12k **(0.812, +14.2%)**. This suggests that scDiVa better captures latent global ranking dependencies under the evaluated re-

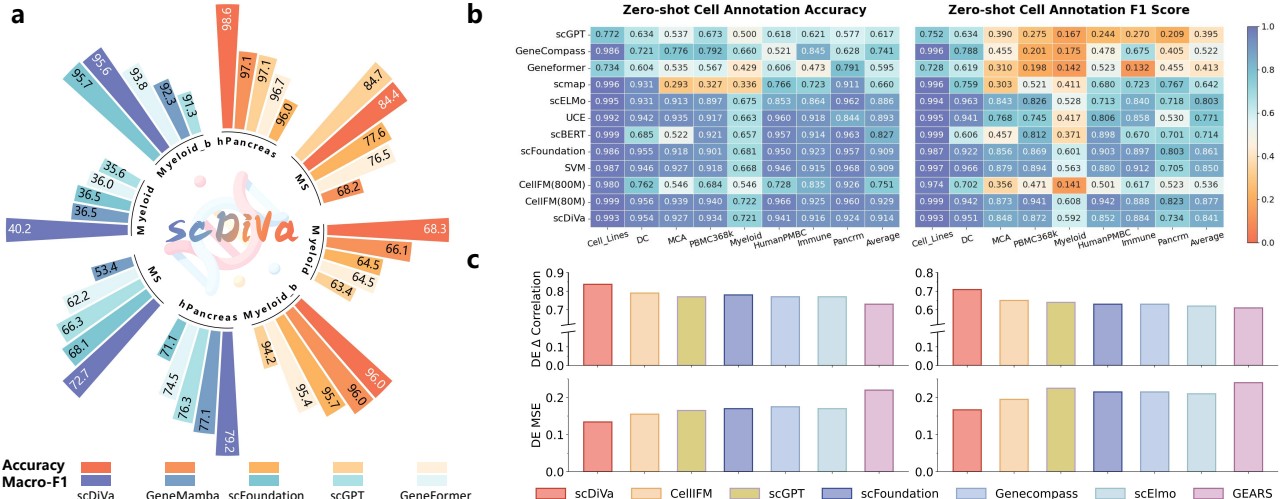

*Figure 4.* **Comprehensive evaluation of scDiVa performance.** (a) Cross-batch fine-tuning performance on hPancreas, MS, Myeloid, and Myeloid_b, with complete numerical results provided in Appendix Table 8. (b) Zero-shot cell annotation performance obtained by freezing the backbone and training only the MLP head. (c) Perturbation prediction comparison on Adamson and Norman using standard DE metrics. The full benchmark with centered and DE-aware metrics is reported in Table 3.

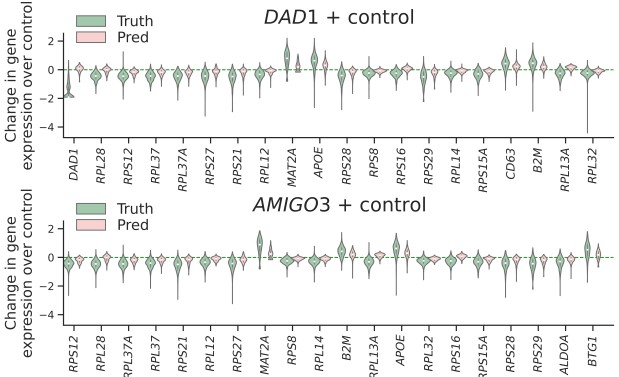

*Figure 5.* **Predicted vs. observed expression shifts in the Adamson dataset.** Distributional changes for the top 20 differentially expressed genes under DAD1 and AMIGO3 perturbations.

construction setting. Further analysis on the intersection of reconstructed gene features across datasets is presented in Figure 8 in Appendix F.1.

To investigate robustness, Figure 2 compares One-step Inference with Multi-step Diffusion. While one-step denoising provides strong reconstruction fidelity, a 30% masking setting induces slight distribution shifts, with local rank deviations and heavy-tailed residuals indicating limitations in handling uncertainty. Conversely, multi-step diffusion acts as an iterative denoising mechanism, pulling samples toward low-energy manifolds to improve Rank convergence and reduce residual variance. However, given the significant latency cost for primarily local gains, we prioritize computational efficiency. As one-step denoising provides sufficient fidelity, we adopt it as the default strategy for subsequent downstream biological tasks.

## 4.3. Multi-batch Integration

We systematically evaluated scDiVa on complex multi-batch integration tasks, aiming to harmonize datasets from distinct batches (Qi et al., 2023) while eliminating technical effects and preserving intrinsic biological heterogeneity.

To address this, we fine-tune the full scDiVa encoder with a combination of reconstruction, batch-adversarial regularization, and supervised contrastive preservation of biological labels when annotations are available. The batch-adversarial term uses a Gradient Reversal Layer (GRL) to reduce batch-identifiable information in the latent representation, while the supervised contrastive term encourages cells with the same cell-type label to remain consistently close across batches. Full objective definitions and key optimization details are provided in Appendix D.1.

Fine-tuning was performed across five benchmarks: Immune, PBMC12k, BMMC, Perirhinal Cortex, and COVID-19, with performance evaluated via latent embeddings (Figure 3; see Figure 10 in Appendix F.2 for visualizations of the other four datasets).

For rigorous comparison, we used Avg-batch to assess batch correction and Avg-bio to evaluate biological conservation, metrics that reflect a fundamental trade-off between noise removal and feature preservation (Meng et al., 2024).

Table 2 demonstrates that scDiVa effectively balances batch mixing and biological conservation. Compared with the original integration baselines, scDiVa achieves the best Avg-batch on four of five datasets and the best Avg-bio on four of five datasets, with notable gains on BMMC and COVID-19. The expanded comparison with recent foundation-model

baselines provides a more balanced picture. scDiVa remains strongest in Avg-batch on Immune, PBMC12k, BMMC, and COVID-19, while GeneMamba is slightly higher on Perirhinal Cortex. For Avg-bio, scDiVa is strongest on BMMC, Perirhinal Cortex, and COVID-19, while GeneMamba and CellFM perform best on Immune and PBMC12k, respectively. These results indicate that scDiVa is not uniformly dominant across all integration settings, but remains highly competitive and often strongest under both batch-mixing and biological-conservation criteria.

### 4.4. Cell Type Annotation

We evaluate cellular representations through two paradigms across supervised and transfer regimes: fine-tuning for cross-batch adaptation and zero-shot evaluation for intrinsic linear separability using only an MLP head.

In fine-tuning settings (Figure 4a), scDiVa achieves competitive Accuracy and imbalance-sensitive Macro-F1, with full comparisons reported in Appendix Table 8. On hPancreas, scDiVa reaches 0.986 Accuracy and 0.7919 Macro-F1, indicating discriminative representations. On the highly imbalanced MS dataset, it attains 0.7271 Macro-F1, improving over GeneMamba's 0.5342 by 36%. Across the expanded appendix comparison, scDiVa obtains the best Macro-F1 on two of four datasets and the best or near-best Accuracy on three of four datasets, suggesting improved decision boundaries for rare populations while remaining competitive with recent foundation-model baselines. Detailed confusion matrices and clustering heatmaps are shown in Figure 7 (Appendix F.3).

Zero-shot evaluation across seven datasets (Figure 4b) suggests broad semantic alignment under minimal task-specific supervision. scDiVa achieves an average accuracy of 0.914 and Macro-F1 of 0.841, outperforming Transformer baselines such as scGPT and Geneformer while remaining competitive with specialized models such as CellFM. Together, the fine-tuning and zero-shot results suggest that the learned latent space provides reliable initial classification while retaining strong adaptation potential for downstream tasks with limited supervision.

### 4.5. Perturbation Prediction

Gene perturbation prediction evaluates expression shifts under held-out genetic interventions, focusing on perturbation-level generalization rather than cross-cell-line transfer. Since standard global-fit metrics can be affected by shared control-to-perturbation shifts and may overestimate target-specific reasoning (Viñas Torné et al., 2025; Mejia et al., 2025), we report both conventional DE metrics and stricter centered or DE-aware metrics (Viñas Torné et al., 2025; Wu et al., 2025b; 2026). Perturbation identities are encoded with a dedicated [PERT] token, and the encoder is fine-tuned us-

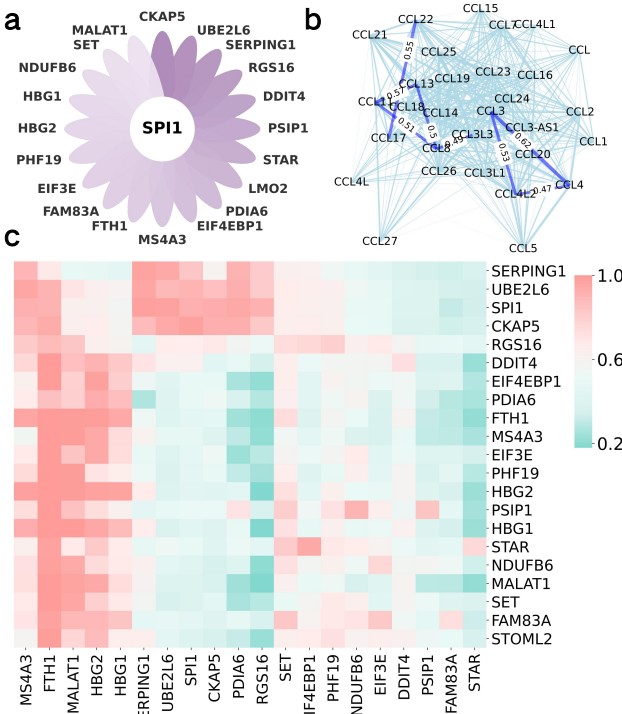

*Figure 6.* **Attention-derived regulatory hypothesis analysis.** (a) Global gene association network constructed from scDiVa attention statistics. (b) Top-ranked SPI1 neighborhood used for regulatory hypothesis generation. (c) Attention heatmap illustrating the hub-like structure of SPI1 within the learned association pattern.

ing signal-sensitive weighted MSE and a DE-aware ranking term. We evaluate Adamson with 87 single-gene perturbations and Norman with 131 double-gene perturbations. Baselines include GEARS, CellFM, scGPT, Geneformer, UCE, scBERT, scELMO (Liu et al., 2026), GeneCompass, scFoundation, and simple Mean, Linear, Additive, and Nochange controls when applicable (Ahlmann-Eltze et al., 2025; Wong et al., 2025; Wu et al., 2025b). All methods use aligned preprocessing, perturbation-level splits, gene sets, pseudobulk aggregation, metrics, and model selection. For Norman, Additive is used as the primary simple baseline because it directly matches the combinatorial setting, while Mean and Linear are reported for Adamson.

As shown in Table 3, scDiVa performs best on Adamson across standard, centered, and rank-based metrics, including C-DEP, DE-AUPRC, LFCSpear, and HR@20. On Norman, Additive remains exceptionally strong, consistent with recent findings that simple baselines can be highly competitive in combinatorial perturbation prediction (Ahlmann-Eltze et al., 2025; Wong et al., 2025; Wu et al., 2025b). Among learned models, scDiVa achieves the best C-DEP and LFCSpear while remaining competitive on DE-AUPRC and HR@20. Overall, scDiVa improves learned perturbation modeling under centered and DE-aware evaluation, but simple baselines remain essential controls and these metrics

*Table 3.* **Perturbation benchmark under aligned protocols.** Each cell reports **Adamson / Norman**. C-DEP denotes centered DE Pearson, AUPRC denotes DE-AUPRC, LFCSp denotes LFCSpear, and HR@20 denotes HitRate@20. The best , second-best , and third-best results are highlighted within each dataset and metric.

| Method | Pearson$\Delta_{de}$ ↑ | DEMSE ↓ | DEP ↑ | C-DEP ↑ | AUPRC ↑ | LFCSp ↑ | HR@20 ↑ |
|---|---|---|---|---|---|---|---|
| *Cell format: Adamson / Norman. Missing entries are denoted by –.* | | | | | | | |
| Mean | 0.801 / – | 0.239 / – | 0.742 / – | 0.158 / – | 0.258 / – | 0.341 / – | 0.319 / – |
| Linear | 0.821 / – | 0.210 / – | 0.771 / – | 0.194 / – | 0.294 / – | 0.389 / – | 0.356 / – |
| Additive | – / **0.932** | – / **0.080** | – / **0.781** | – / 0.269 | – / **0.348** | – / 0.423 | – / **0.461** |
| Nochange | – / – | – / 0.382 | – / – | – / – | – / 0.081 | – / – | – / 0.098 |
| GEARS | 0.810 / 0.810 | 0.225 / 0.267 | 0.812 / 0.681 | 0.271 / 0.188 | 0.341 / 0.268 | 0.449 / 0.371 | 0.432 / 0.334 |
| CellFM | 0.819 / 0.841 | 0.157 / 0.194 | 0.821 / 0.703 | 0.289 / 0.212 | 0.362 / 0.291 | 0.471 / 0.391 | 0.458 / 0.372 |
| scBERT | 0.790 / 0.791 | 0.250 / 0.291 | 0.778 / 0.664 | 0.191 / 0.174 | 0.279 / 0.243 | 0.369 / 0.334 | 0.346 / 0.304 |
| Geneformer | 0.811 / 0.880 | 0.231 / 0.124 | 0.796 / 0.748 | 0.228 / 0.236 | 0.312 / 0.321 | 0.414 / 0.432 | 0.392 / 0.401 |
| UCE | 0.831 / 0.790 | 0.193 / 0.286 | 0.804 / 0.672 | 0.247 / 0.177 | 0.334 / 0.251 | 0.435 / 0.343 | 0.413 / 0.313 |
| scGPT | 0.698 / 0.762 | 0.169 / 0.232 | 0.691 / 0.638 | 0.108 / 0.147 | 0.256 / 0.222 | 0.309 / 0.301 | 0.287 / 0.272 |
| scELMO | 0.798 / 0.799 | 0.171 / 0.211 | 0.782 / 0.676 | 0.206 / 0.191 | 0.292 / 0.262 | 0.381 / 0.361 | 0.359 / 0.333 |
| GeneCompass | 0.771 / 0.808 | 0.182 / 0.222 | 0.763 / 0.689 | 0.176 / 0.201 | 0.274 / 0.279 | 0.356 / 0.384 | 0.334 / 0.351 |
| scFoundation | 0.808 / 0.769 | 0.177 / 0.221 | 0.793 / 0.649 | 0.224 / 0.159 | 0.304 / 0.233 | 0.397 / 0.312 | 0.375 / 0.284 |
| **scDiVa** | **0.838** / 0.861 | **0.135** / 0.163 | **0.842** / 0.724 | **0.337** / **0.271** | **0.421** / 0.341 | **0.543** / **0.433** | **0.529** / 0.441 |

alone do not establish causal mechanism recovery. Qualitative expression-shift visualizations are shown in Figure 5, with additional top-$k$ analyses in Appendix F.4.

### 4.6. Gene Regulatory Network Inference and Hypothesis Generation

To examine whether scDiVa captures biologically meaningful regulatory structure, we construct attention-derived gene association networks as hypothesis-generating summaries rather than definitive causal graphs. A global association network is shown in Figure 6a, with extended family-level visualizations in Appendix Figure 9. We focus on SPI1, a myeloid regulator, and compare its top-ranked scDiVa neighbors against curated SPI1 targets under a matched background gene set. This design controls for network-size effects and avoids inflating enrichment because many genes are present, making the result conservative. This also makes the analysis less sensitive to degree heterogeneity. Among the top-20 SPI1 neighbors, 4 overlap with curated targets, yielding an odds ratio of 10.6, Fisher exact test $p = 1.1 \times 10^{-3}$, and empirical $p = 0.006$ against a random baseline (Appendix Table 9). The SPI1 neighborhood includes genes consistent with myeloid and immune-associated programs, suggesting that scDiVa attention patterns can prioritize plausible regulatory candidates. Because attention scores may reflect indirect, context-dependent, or non-regulatory associations, we interpret these results as statistically contextualized regulatory hypotheses rather than evidence of direct causal regulation.

## 5. Conclusion

In this work, we present scDiVa, a foundation model for single-cell representation learning based on masked discrete diffusion. By aligning the generative process with dropout-like gene non-detection and adopting a dual denoising objective, scDiVa mitigates the structural biases of autoregressive models and enables accurate reconstruction of both gene identity and expression magnitude. Evaluations across integration, annotation, and perturbation tasks suggest that modeling gene expression with bidirectional masked denoising can improve representation quality and biological fidelity under the evaluated protocols. These results highlight discrete diffusion as a principled alternative to sequential paradigms for large-scale single-cell modeling.

## Code and Data Availability

We release the scDiVa weights and checkpoints at `https://github.com/wangmingxuan666/ScDiVa`, and provide the downstream datasets used in our evaluation at `https://huggingface.co/warming666/ScDiVa`. Following a model-weight release protocol, the repository provides trained weights rather than a training codebase; due to company policy, confidentiality constraints, and data governance restrictions, the full training code, vocabulary construction pipeline, and proprietary pre-training corpus cannot be publicly released at this stage, although all downstream evaluations are conducted on publicly available datasets and we are working toward a complete open-source release in the future.

## Acknowledgements

We gratefully acknowledge the support of Public Computing Cloud, Renmin University of China. This work was also supported by the fund for building world-class universities (disciplines) of Renmin University of China.

We are grateful to the Area Chairs and the anonymous reviewers for their constructive comments, fair evaluation, and thoughtful assessment of our work.

I would like to thank Advisor Ma for his invaluable guidance and continuous support throughout my first work.

During the development of this paper, I experienced a major life event; I am deeply grateful to my late grandfather, who always cared about my research, and whose blessings from heaven I will always cherish.

## Impact Statement

This paper presents scDiVa, a foundation model for computational biology and single-cell genomics. The work aims to improve cellular representation learning, with potential benefits for understanding disease mechanisms, drug target discovery, and precision medicine.

This work may also raise societal and ethical considerations, including the computational cost of large-scale model training and privacy concerns in future clinical applications. Although our downstream evaluations use public anonymized datasets, the proprietary pre-training corpus is handled under institutional data-governance and privacy constraints. Predictions from scDiVa, including perturbation responses and attention-derived regulatory associations, should be interpreted as computational hypotheses rather than clinical or causal conclusions without experimental validation.

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

# A. Additional Methodological Details

## A.1. Biological Interpretation of Masked Discrete Diffusion

Single-cell RNA sequencing measurements can be viewed as noisy observations of an underlying biological state, where gene expression signals are subject to stochastic loss due to limited capture efficiency, amplification bias, and sequencing depth constraints. In this context, the masked discrete diffusion process employed by scDiVa provides a natural abstraction of technical dropout.

Unlike continuous Gaussian noise, which perturbs values symmetrically in Euclidean space, dropout events correspond to the complete disappearance of gene-level signals. The absorbing [MASK] state used in scDiVa explicitly models this irreversible information loss. As diffusion time increases, the probability of observing a true gene signal decreases monotonically, reflecting progressively lower effective sequencing depth. This correspondence grounds the forward diffusion process in the physical data-generating mechanism of single-cell experiments.

## A.2. Absorbing-State Corruption and Dropout Correspondence

The forward corruption process in scDiVa independently replaces each gene token with an absorbing [MASK] state with probability proportional to diffusion time. This design differs fundamentally from additive noise models, where corrupted values remain informative through magnitude. In contrast, masked corruption enforces a discrete transition between observable and unobservable states, matching the binary nature of dropout events.

Importantly, the absorbing state ensures that once a gene is masked, no residual information about its original identity or magnitude remains in the corrupted representation. This property allows the reverse denoising process to learn explicit conditional dependencies among genes, rather than relying on smooth interpolation in value space.

## A.3. Comparison with Autoregressive and Gaussian Diffusion Models

Autoregressive (AR) models factorize the joint distribution over genes into a product of conditional distributions ordered by an arbitrary sequence. This ordering introduces artificial causal dependencies that are biologically implausible for gene regulatory networks, where interactions are typically symmetric or cyclic. Moreover, AR generation is susceptible to exposure bias, as early prediction errors propagate irreversibly.

Gaussian diffusion models avoid explicit ordering but impose an ordinal structure on gene expression values, as-suming that small perturbations preserve semantic meaning. This assumption is problematic for discrete gene activation events and sparse count distributions. By contrast, masked discrete diffusion directly models presence or absence through state transitions and performs reconstruction using full bidirectional context, avoiding both ordering bias and ordinal assumptions.

## A.4. Motivation for Entropy-Normalized Serialization

Gene expression distributions exhibit extreme imbalance: a small number of housekeeping genes are ubiquitously expressed across cell types, while many low-frequency genes carry high discriminative value. Ranking genes solely by expression magnitude therefore biases token allocation toward high-abundance but low-information features.

Entropy-normalized serialization mitigates this effect by weighting expression values with inverse population-level entropy. Genes that are consistently expressed across cells receive lower priority, while genes with high cell-type specificity are emphasized. This strategy allows a finite token budget to encode maximal discriminative information and improves downstream transfer performance under fixed context length constraints.

## A.5. Inductive Bias of Transformer Parameterization

ScDiVa employs a bidirectional Transformer encoder to parameterize the reverse denoising distribution. Absolute positional encodings are omitted because gene order in the input sequence is not biologically meaningful. Introducing absolute positions would impose a Cartesian structure that is invariant neither to permutation nor to feature selection.

Relative positional encoding via RoPE is retained to allow the model to learn structured dependencies among co-occurring genes within the serialized sequence. Since serialization is deterministic given the ranking rule, relative positions encode comparative importance rather than temporal order. This design reduces autoregressive ordering bias while retaining sufficient inductive bias to learn structured gene-gene interactions.

## A.6. Latent Anchor Token and High-Mask Stability

At high diffusion times, a large fraction of gene tokens are masked, resulting in sparse and unstable inputs. To address this, scDiVa introduces a latent anchor token [LAT] that participates in self-attention but is never masked. This token aggregates global information from observed genes and serves as a persistent conditioning signal during denoising.

Empirically, the latent anchor improves training stability and prevents identity drift when the visible token set is small. Conceptually, [LAT] functions as a global summary of the

partially observed cell state, enabling consistent reconstruction even under extreme corruption.

## A.7. Depth-Robust Sampling under Sparse Observations

Uniform sampling of diffusion time exposes the model to varying corruption levels. We interpret this as a robustness strategy for sparse observations, not as an exact simulator of sequencing depth. In real single-cell measurements, lower sequencing depth changes library size and count statistics globally, while gene-token masking models the non-detection of individual gene events. These mechanisms are related through dropout-like missingness, but they are not equivalent.

In scDiVa, low diffusion times preserve most observed gene tokens, whereas high diffusion times create partially observed profiles with increased non-detection. Training across this spectrum encourages the model to recover gene identity and expression magnitude from incomplete observations. This improves robustness under sparse measurements, while explicit normalization and task-specific adaptation may still be required for downstream settings with strong technical batch effects.

## A.8. Formal Correspondence between Absorbing Masking and Technical Dropout

**Forward process.** Let a serialized cell be represented by a length-$L$ sequence of discrete gene identity tokens $x_0 = (x_0^1, \ldots, x_0^L)$ with $x_0^i \in \mathcal{V}$, where $\mathcal{V}$ is a finite vocabulary of genes. Let $\varnothing$ denote the absorbing [MASK] state. The (continuous-time) masked discrete diffusion defines, for each position $i$ and time $t \in [0, 1]$,

$$q(x_t^i \mid x_0^i) = (1 - t)\delta(x_t^i, x_0^i) + t\delta(x_t^i, \varnothing), \quad (5)$$

where $\delta(\cdot, \cdot)$ is the Kronecker delta on $\mathcal{V} \cup \varnothing$. Assuming conditional independence across positions yields $q(x_t \mid x_0) = \prod_{i=1}^{L} q(x_t^i \mid x_0^i)$.

**Theorem A.1** (Dropout–Diffusion Isomorphism). *Define an observation map $\phi : \mathcal{V} \cup \varnothing \to \mathcal{V} \cup 0$ by $\phi(g) = g$ for $g \in \mathcal{V}$ and $\phi(\varnothing) = 0$ (interpreting [MASK] as complete information loss). For each position $i$, define the observed "signal" random variable $y_t^i = \phi(x_t^i)$. Then conditioned on $x_0^i$, $y_t^i$ follows a* zero-inflated *(dropout) corruption:*

$$q(y_t^i \mid x_0^i) = (1 - t)\delta(y_t^i, x_0^i) + t\delta(y_t^i, 0). \quad (6)$$

*In particular, as $t \to 1$, $q(y_t^i \mid x_0^i) \Rightarrow \delta(y_t^i, 0)$, i.e., complete dropout.*

*Proof.* By definition of $\phi$ and the forward kernel in Eq. (5),

$$\mathbb{P}(y_t^i = x_0^i \mid x_0^i) = \mathbb{P}(x_t^i = x_0^i \mid x_0^i) = 1 - t, \quad (7)$$
$$\mathbb{P}(y_t^i = 0 \mid x_0^i) = \mathbb{P}(x_t^i = \varnothing \mid x_0^i) = t. \quad (8)$$

Since $y_t^i$ takes values only in $x_0^i, 0$ under $\phi$, the conditional distribution is exactly the mixture in Eq. (6). Taking $t \to 1$ gives $\mathbb{P}(y_t^i = 0 \mid x_0^i) \to 1$, hence weak convergence to $\delta(y_t^i, 0)$. $\square$

**Connection to expression dropout.** For a paired gene–value representation $(g_i, v_i)$, define the value corruption $\tilde{v}_t^i = \mathbb{I}[x_t^i \neq \varnothing] \cdot v_i$. Then $\tilde{v}_t^i$ obeys the same zero-inflated form:

$$q(\tilde{v}_t^i \mid v_i) = (1 - t)\delta(\tilde{v}_t^i, v_i) + t\delta(\tilde{v}_t^i, 0), \quad (9)$$

which matches the canonical abstraction of technical dropout as complete signal disappearance.

## A.9. ELBO Derivation for Masked Discrete Diffusion

We provide a rigorous likelihood lower bound for the discrete identity component; the value regression term is then interpreted as a Gaussian log-likelihood term.

**Discrete-time construction.** Let $0 = t_0 < t_1 < \cdots < t_K = 1$ be a discretization. Define the forward Markov chain

$$q(x_{t_k} \mid x_{t_{k-1}}) = \prod_{i=1}^{L} q(x_{t_k}^i \mid x_{t_{k-1}}^i),$$
$$q(x_{t_k}^i = \varnothing \mid x_{t_{k-1}}^i \neq \varnothing) = \frac{t_k - t_{k-1}}{1 - t_{k-1}}, \quad q(\varnothing \mid \varnothing) = 1,$$
$$(10)$$

so that the marginal $q(x_{t_k}^i \mid x_0^i)$ equals Eq. (5) at $t = t_k$. Let the prior at $t_K$ be the fully-masked absorbing state:

$$p(x_{t_K}) = \delta(x_{t_K}, \varnothing^L). \quad (11)$$

Let the reverse model be a parametric family $p_\theta(x_{t_{k-1}} \mid x_{t_k})$ (implemented by a bidirectional Transformer conditioned on $x_{t_k}$ and $t_k$).

**Model likelihood.** The induced model distribution over clean sequences is

$$p_\theta(x_0) = \sum_{x_{t_1}, \ldots, x_{t_K}} p(x_{t_K}) \prod_{k=K}^{1} p_\theta(x_{t_{k-1}} \mid x_{t_k}), \quad (12)$$

where $x_{t_0} \equiv x_0$.

**Variational lower bound (ELBO).** Choose the variational posterior as the forward diffusion path $q(x_{t_{1:K}} \mid$

$x_0) = \prod_{k=1}^{K} q(x_{t_k} \mid x_{t_{k-1}})$. Then,

$$\log p_\theta(x_0) = \log \mathbb{E}_{q(x_{t_{1:K}}|x_0)} \left[ \frac{p(x_{t_K}) \prod_{k=K}^{1} p_\theta(x_{t_{k-1}} \mid x_{t_k})}{q(x_{t_{1:K}} \mid x_0)} \right]$$

$$\geq \mathbb{E}_q \left[ \log p(x_{t_K}) + \sum_{k=K}^{1} \log \frac{p_\theta(x_{t_{k-1}}|x_{t_k})}{q(x_{t_k}|x_{t_{k-1}})} \right]$$

$$\triangleq \mathcal{L}_{\mathrm{ELBO}}(\theta; x_0), \tag{13}$$

where the inequality is Jensen's inequality. By construction, $\mathcal{L}_{\mathrm{ELBO}}(\theta; x_0) \leq \log p_\theta(x_0)$.

**Optimization target.** In Eq. (13), the terms $\log p(x_{t_K})$ and $\log q(\cdot)$ do not depend on $\theta$. Therefore, maximizing $\mathcal{L}_{\mathrm{ELBO}}$ is equivalent to maximizing $\mathbb{E}_q \left[ \sum_{k=K}^{1} \log p_\theta(x_{t_{k-1}} \mid x_{t_k}) \right]$. A standard Monte-Carlo estimator is obtained by sampling a random timestep $k$ (or continuous $t$) and training the model to predict the uncorrupted token identities at masked positions.

**Single-step objective and the scDiVa loss.** Let $t \sim \mathrm{Unif}(0,1)$ and $x_t \sim q(\cdot \mid x_0, t)$ as in Eq. (5). Let $M_t = \{i : x_t^i = \varnothing\}$ be the masked index set. Assume the reverse conditional factorizes over positions given $x_t$ (standard in masked modeling):

$$p_\theta(x_0 \mid x_t, t) = \prod_{i \in M_t} p_\theta(x_0^i \mid x_t, t), \tag{14}$$

$$\log p_\theta(x_0 \mid x_t, t) = \sum_{i \in M_t} \log p_\theta(x_0^i \mid x_t, t). \tag{15}$$

Using the practical normalization $|M_t|^{-1}$ (equivalently $\approx (tL)^{-1}$ in expectation) yields the training objective

$$\mathcal{L}_{\mathrm{id}}(\theta) \triangleq \mathbb{E}_{x_0, t, x_t} \left[ \frac{1}{|M_t|} \sum_{i \in M_t} \log p_\theta(x_0^i \mid x_t, t) \right], \tag{16}$$

which is a (constant-shifted) stochastic estimator of the reverse term in Eq. (13), and thus maximizes a lower bound on $\mathbb{E}_{p_{\mathrm{data}}}[\log p_\theta(x_0)]$.

**Incorporating continuous values.** For each token position $i$, scDiVa predicts both identity and value. Let $g_i$ denote the true gene identity and $v_i \in \mathbb{R}$ its (log-normalized) expression. We interpret the regression term as a Gaussian likelihood $p_\theta(v_i \mid x_t, t) = \mathcal{N}(\hat{v}_i, \sigma^2)$ with fixed variance $\sigma^2$. Then

$$\log p_\theta(v_i \mid x_t, t) = -\frac{1}{2\sigma^2} |\hat{v}_i - v_i|^2 + \mathrm{const.} \tag{17}$$

Consequently, defining $\lambda \triangleq \frac{1}{2\sigma^2}$, the joint objective

$$\mathcal{L}_{\mathrm{total}}(\theta) \triangleq \mathcal{L}_{\mathrm{id}}(\theta) + \lambda \mathbb{E} \left[ \frac{1}{|M_t|} \sum_{i \in M_t} \log p_\theta(v_i \mid x_t, t) \right] \tag{18}$$

maximizes a lower bound on the joint log-likelihood of identities and values under the assumed factorization.

## A.10. Algorithms

---

**Algorithm 1** scDiVa Training

---

**Require:** Dataset $\mathcal{D}$ of cells; max length $L_{\max}$; model $p_\theta$ with `[LAT]`; loss weight $\lambda$
1: Initialize $\theta$ (AdamW optimizer)
2: **repeat**
3:     Sample mini-batch of cells $(g_{1:L_b}^{(b)}, v_{1:L_b}^{(b)})_{b=1}^{B}$ after entropy-normalized serialization
4:     Pad to $L_{\max}$ using `[PAD]`; prepend `[LAT]` token (never masked)
5:     Sample diffusion time $t \sim \mathrm{Unif}(0,1)$
6:     **for** $b = 1$ to $B$ **do**
7:         For each position $i$ (excluding `[LAT]` and `[PAD]`), sample mask $m_i \sim \mathrm{Bernoulli}(t)$
8:         Set corrupted token $x_t^i \leftarrow \varnothing$ if $m_i = 1$, else $x_t^i \leftarrow g_i$; record $M_t = i : m_i = 1$
9:         Replace masked value input with a sentinel (e.g., 0) or a learned `[MASK]` value embedding
10:     **end for**
11:     Forward pass: obtain hidden states $h_i$ from the bidirectional Transformer
12:     Predict gene logits $\hat{y}_{\mathrm{id}}^i$ and value predictions $\hat{v}_i$ for all positions
13:     Compute identity loss on masked positions:
$$\mathcal{L}_{\mathrm{id}} \leftarrow -\frac{1}{|M_t|} \sum_{i \in M_t} \log p_\theta(g_i \mid x_t, t)$$
14:     Compute value loss on masked positions:
$$\mathcal{L}_{\mathrm{val}} \leftarrow \frac{1}{|M_t|} \sum_{i \in M_t} |\hat{v}_i - v_i|^2$$
15:     Total loss: $\mathcal{L} \leftarrow \mathcal{L}_{\mathrm{id}} + \lambda \mathcal{L}_{\mathrm{val}}$
16:     Update parameters $\theta \leftarrow \mathrm{AdamW}(\theta, \nabla_\theta \mathcal{L})$
17: **until** convergence

---

## B. Dataset and Preprocessing Details

### B.1. Entropy-Normalized Serialization

Let $v_{c,g} \in \mathbb{R}_{\geq 0}$ denote the (log-normalized) expression of gene $g$ in cell $c$. To quantify population-level ubiquity, we compute Shannon entropy for each gene $g$. Let $X_g$ be a discretized random variable obtained by binning $v_{c,g}$ across the pre-training corpus (e.g., via fixed-width or quantile bins), with empirical probabilities $p_g(x) = \mathbb{P}(X_g = x)$.

**Algorithm 2** scDiVa Inference

---

**Require:** Trained model $p_\theta$; length $L$; steps $K$; schedule $\{t_k\}_{k=0}^{K}$ with $t_0 = 0 < t_1 < \cdots < t_K = 1$

1: Initialize $x_{t_K} \leftarrow \varnothing^L$ (fully masked), prepend `[LAT]`
2: **for** $k = K$ down to 1 **do**
3:     Run mask predictor to obtain $p_\theta(\cdot \mid x_{t_k}, t_k)$ and value predictions $\hat{v}$
4:     For each masked position $i$, sample $\tilde{g}_i \sim p_\theta(\cdot \mid x_{t_k}, t_k)$ and set provisional unmasking
5:     Determine target masking ratio $t_{k-1}$; unmask a fraction $(t_k - t_{k-1})/t_k$ of currently masked tokens
6:     (Optional) Low-confidence remasking: remask the fraction $t_{k-1}/t_k$ of tokens with lowest confidence to match $t_{k-1}$
7:     Set $x_{t_{k-1}}$ accordingly; keep `[LAT]` unchanged
8: **end for**

---

Then the gene entropy is

$$H(g) = -\sum_x p_g(x) \log\left(p_g(x)\right). \tag{19}$$

Given a cell-specific value $v_{c,g}$, we define the entropy-normalized ranking score

$$S_g(c) = \frac{v_{c,g}}{H(g) + \epsilon}, \tag{20}$$

where $\epsilon > 0$ prevents division by zero. For each cell $c$, scDiVa selects the top-$L_{\max}$ genes by descending $S_g(c)$ and serializes them deterministically into a length-$L_{\max}$ sequence.

### B.2. Dataset Composition and Preprocessing Details

The pre-training corpus comprises 59,162,450 single-cell transcriptomes aggregated from diverse tissues, conditions, and sequencing technologies. Cells with fewer than 200 detected genes were removed. Expression counts were log-normalized following standard preprocessing practice.

Entropy statistics were computed globally across the pre-training corpus and fixed prior to model training. For each cell, the top 1,200 genes ranked by entropy-normalized score were selected and serialized deterministically. This identical preprocessing pipeline was applied during pre-training and downstream evaluation to ensure distributional consistency.

### B.3. Pre-training Data Summary

The pre-training corpus consists of a large-scale, proprietary single-cell transcriptomic dataset aggregated from internal sources. Due to strict data privacy regulations and commercial confidentiality agreements, the specific composition, donor metadata, and source breakdown of this corpus cannot be publicly disclosed. However, the dataset is curated to ensure high diversity, covering a wide range of tissue types, developmental stages, and sequencing technologies comparable to major public archives. Due to company policy and data-governance restrictions, donor-level metadata, source-level composition, exact tissue proportions, and sequencing-platform proportions of the proprietary pre-training corpus cannot be disclosed. We therefore report only the total number of cells, the unified preprocessing protocol, and the downstream public datasets used for evaluation.

### B.4. Downstream Dataset Statistics

For all downstream evaluation tasks, including fine-tuning, zero-shot learning, and integration, we exclusively utilized publicly available, open-source datasets. These benchmarks (summarized in Table 4) represent standard community resources, ensuring the transparency and reproducibility of our evaluation metrics.

## C. Model Architecture and Implementation Details

### C.1. Joint Embedding Formulation

Each serialized position corresponds to a gene identity $g \in \mathcal{V}$ and a continuous value $v \in \mathbb{R}$. Let $d$ be the hidden dimension. The input representation is

$$h_{\text{input}} = \text{Emb}_{\text{id}}(g) + \text{MLP}_{\text{val}}(v), \tag{21}$$

where $\text{Emb}_{\text{id}} : \mathcal{V} \to \mathbb{R}^d$ is a learnable embedding table, and $\text{MLP}_{\text{val}} : \mathbb{R} \to \mathbb{R}^d$ is a 2-layer perceptron projecting scalar values to $\mathbb{R}^d$, e.g.,

$$\text{MLP}_{\text{val}}(v) = W_2 \sigma(W_1 v + b_1) + b_2, \tag{22}$$

with nonlinearity $\sigma(\cdot)$ (e.g., SiLU).

### C.2. Pre-Norm Transformer Block Dynamics

Let $x_l \in \mathbb{R}^{L \times d}$ denote the sequence representation at layer $l$. Using Pre-Norm with RMSNorm, the $l$-th block computes

$$x_l' = x_l + \text{Attention}\left(\text{RMSNorm}(x_l)\right), \tag{23}$$

$$x_{l+1} = x_l' + \text{FFN}\left(\text{RMSNorm}(x_l')\right). \tag{24}$$

No causal mask is used, enabling bidirectional conditioning across the serialized gene sequence.

### C.3. Component Formulations

**Multi-head attention.** For head dimension $d_h = d/H$ with $H$ heads, define $Q = XW_Q, K = XW_K, V = XW_V$. RoPE is applied to $(Q, K)$ (below). The attention output is

$$\text{Attention}(X) = \text{Concat}\left(\text{head}_1, \ldots, \text{head}_H\right) W_O,$$

$$\text{head}_h = \text{softmax}\left(\frac{\tilde{Q}_h \tilde{K}_h^\top}{\sqrt{d_h}}\right) \tilde{V}_h. \tag{25}$$

*Table 4.* Downstream dataset statistics used in our evaluations. Sparsity is the fraction of zero entries in the raw gene-by-cell count matrix.

| Dataset | Task | N_Cells | N_Genes | Sparsity | Batches | CellTypes |
|---|---|---|---|---|---|---|
| Immune | Gene Reconstruction / GRN | 32,484 | 12,303 | 88.15% | 9 | 16 |
| Zheng68k | Gene Reconstruction / GRN | 68,579 | 32,738 | 98.34% | *N/A* | *N/A* |
| BMMC | Multi-batch Integration | 90,261 | 14,087 | 88.87% | 12 | 45 |
| Perirhinal | Multi-batch Integration | 17,535 | 59,357 | 96.33% | 2 | 10 |
| PBMC12k | Multi-batch Integration | 11,990 | 3,346 | 86.32% | 2 | 9 |
| COVID-19 | Multi-batch Integration | 20,000 | 1,200 | 89.52% | 2 | 39 |
| MS | Cell Annotation (FT) | 21,312 | 3,000 | 89.28% | 2 | 18 |
| hPancreas | Cell Annotation (FT) | 14,818 | 3,000 | 87.06% | 2 | 14 |
| Myeloid | Cell Annotation (FT) | 13,178 | 3,000 | 80.84% | 2 | 21 |
| Myeloid_b | Cell Annotation (FT) | 9,926 | 3,000 | 81.16% | 2 | 7 |
| Cell Lines | Cell Annotation (Zero-shot) | 9,531 | 32,738 | 89.80% | 3 | 2 |
| DC | Cell Annotation (Zero-shot) | 576 | 26,593 | 80.98% | 2 | 4 |
| HumanPBMC | Cell Annotation (Zero-shot) | 15,476 | 33,694 | 95.20% | 2 | 9 |
| MCA | Cell Annotation (Zero-shot) | 6,954 | 15,006 | 91.22% | 2 | 11 |
| PBMC | Cell Annotation (Zero-shot) | 18,868 | 6,998 | 95.32% | 2 | 7 |
| PBMC_368K | Cell Annotation (Zero-shot) | 4,638 | 14,236 | 94.93% | 2 | 8 |
| Pancrm | Cell Annotation (Zero-shot) | 14,767 | 15,558 | 77.85% | 5 | 15 |
| Adamson | Perturbation Prediction | 68,603 | 5,060 | 79.32% | *N/A* | 1 |
| Norman | Perturbation Prediction | 91,205 | 5,045 | 91.89% | *N/A* | 1 |

**SwiGLU.** The feed-forward network uses SwiGLU gating:

$$\text{SwiGLU}(x) = (\text{SiLU}(xW_g) \odot (xW_u)) W_d. \quad (26)$$

**RMSNorm.** For $\epsilon > 0$ and learnable scale $\gamma \in \mathbb{R}^d$,

$$\text{RMSNorm}(x) = \frac{x}{\sqrt{\text{Mean}(x^2) + \epsilon}} \odot \gamma. \quad (27)$$

**RoPE.** Let $\Theta$ be the RoPE base (we use $\Theta = 10000$). For each head and each position pos, RoPE applies a block-diagonal rotation to pairs of coordinates. For $j = 0, \ldots, \frac{d_h}{2} - 1$, define angular frequency $\omega_j = \Theta^{-2j/d_h}$ and

$$R_{\Theta,\text{pos}}^{(j)} = \begin{bmatrix} \cos(\omega_j, \text{pos}) & -\sin(\omega_j, \text{pos}) \\ \sin(\omega_j, \text{pos}) & \cos(\omega_j, \text{pos}) \end{bmatrix}. \quad (28)$$

Applying $R_{\Theta,\text{pos}}$ to each pair yields $\tilde{Q} = R_{\Theta,\text{pos}} Q$ and $\tilde{K} = R_{\Theta,\text{pos}} K$.

**Dual output heads.** Let $h_L^i \in \mathbb{R}^d$ be the final-layer hidden state at position $i$. scDiVa predicts gene identity logits and a scalar value:

$$\begin{aligned} \hat{y}_{\text{id}}^i &= \text{Linear}_{\text{gene}}(h_L^i) \in \mathbb{R}^{|\mathcal{V}|}, \\ \hat{y}_{\text{val}}^i &= \text{Linear}_{\text{val}}(h_L^i) \in \mathbb{R}. \end{aligned} \quad (29)$$

### C.4. Hyperparameters

Tables 5 and 6 list hyperparameters selected for efficiency. The architecture employs SwiGLU , RMSNorm, and RoPE to ensure stability and expressivity, optimized via AdamW.

*Table 5.* **Model configuration.**

| Item | Value |
|---|---|
| # Layers | 12 |
| Hidden dim $d$ | 512 |
| # Attention heads $H$ | 8 |
| FFN hidden dim | 2048 |
| Vocabulary size $|\mathcal{V}|$ | 41,818 |
| Max sequence length $L_{\max}$ | 1200 |
| Normalization | RMSNorm ($\epsilon = 10^{-5}$) |
| Activation | SwiGLU |
| RoPE base $\Theta$ | 10000 |

*Table 6.* **Training configuration.**

| Item | Value |
|---|---|
| Global batch size | 768 |
| Optimizer | AdamW |
| Loss weight $\lambda$ (value term) | 10.0 |
| Time sampling | $t \sim \text{Unif}(0, 1)$ |

## D. Downstream Adaptation Protocols

We summarize the downstream adaptation protocol for each task in Table 7. Unless otherwise specified, the cell representation is taken from the hidden state of the latent anchor token [LAT]. For reconstruction, the pre-trained identity and value denoising heads are used directly. For integration and perturbation prediction, the full encoder is updated. For annotation, we freeze the first eight Transformer layers and fine-tune the last four layers together with an MLP classifier.

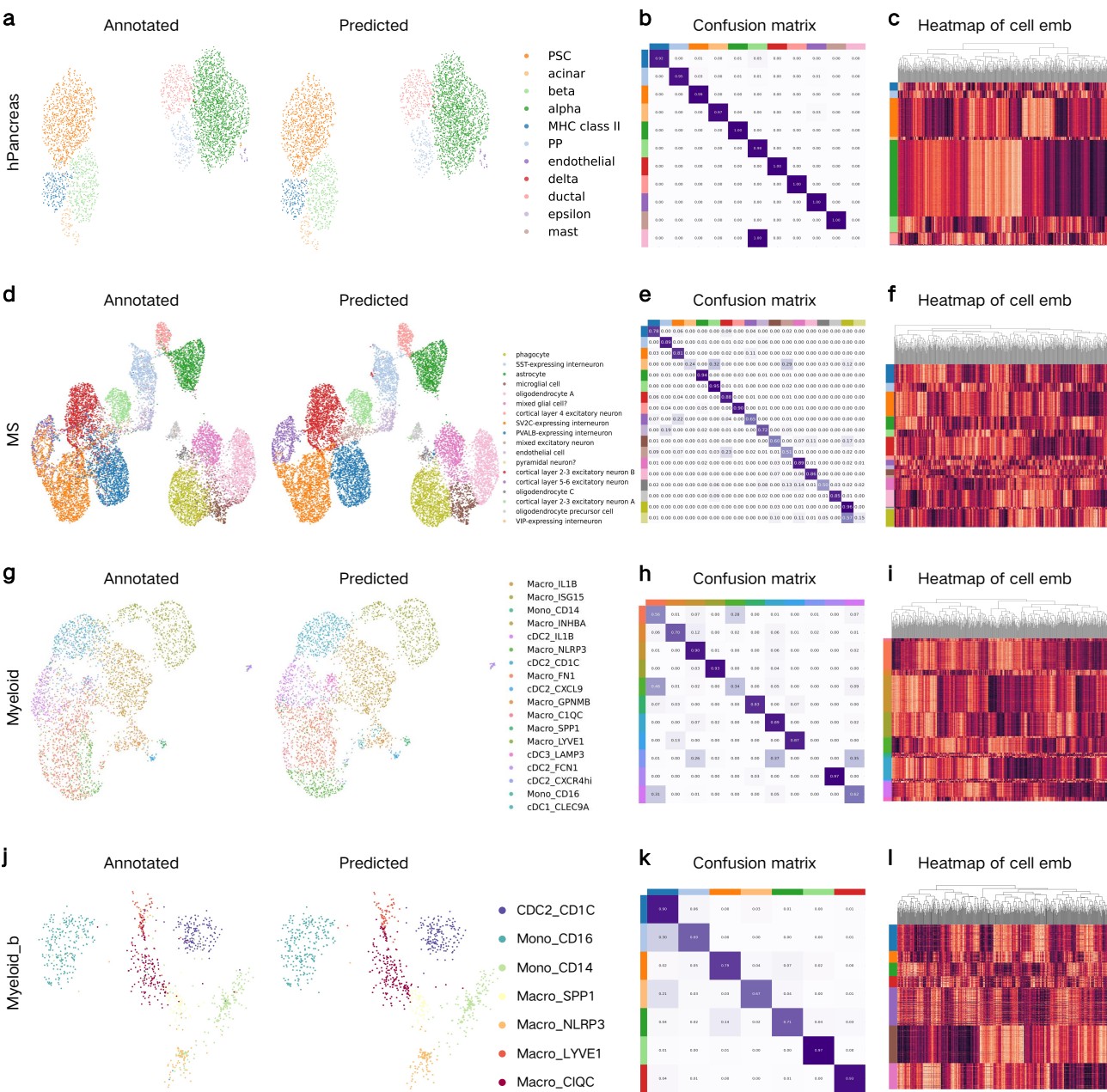

*Figure 7.* **Detailed cell type annotation analysis.** For each dataset, we compare ground-truth annotations with scDiVa predictions, followed by the corresponding confusion matrix and cell-embedding heatmap. Panels (a–c), (d–f), (g–i), and (j–l) show results for hPancreas, MS, Myeloid, and Myeloid_b, respectively. The UMAP visualizations assess global label agreement, the confusion matrices quantify class-wise prediction accuracy, and the heatmaps summarize the structure of learned cell embeddings grouped by predicted cell types. This layout directly enables visual inspection of dataset-specific annotation consistency.

## D.1. Multi-batch Integration Objective

For multi-batch integration, we fine-tune the full sc-DiVa encoder with three objectives: reconstruction, batch-adversarial regularization, and supervised contrastive preservation of biological labels when available. A two-layer batch discriminator $D_{\text{batch}}$ is attached to the cell embedding $h_i$ through a Gradient Reversal Layer (GRL) and is trained to predict the batch label $b_i$:

$$\mathcal{L}_{\text{batch}} = -\sum_i \log D_{\text{batch}}(h_i)_{b_i}. \qquad (30)$$

The discriminator minimizes this objective, while the encoder receives the reversed gradient through the GRL, reducing batch-identifiable information in the latent representation.

When reference cell-type labels are available, we additionally apply supervised contrastive learning to normalized cell embeddings. Positive pairs are cells with the same cell-type label across batches, and negative pairs are cells with different labels:

$$\mathcal{L}_{\text{sup}} = \sum_i \frac{-1}{|P(i)|} \sum_{p \in P(i)} \log \frac{\exp(z_i^\top z_p / \tau)}{\sum_{a \in A(i)} \exp(z_i^\top z_a / \tau)}. \qquad (31)$$

Here, $P(i)$ denotes the set of mini-batch samples sharing the cell-type label of cell $i$, and $A(i)$ denotes the remaining samples in the mini-batch. This term is used only when cell-type annotations are available; therefore, the corresponding protocol is supervised or semi-supervised rather than fully unsupervised.

The full integration objective is:

$$\mathcal{L}_{\text{int}} = \mathcal{L}_{\text{recon}} + \lambda_{\text{adv}} \mathcal{L}_{\text{batch}} + \lambda_{\text{sc}} \mathcal{L}_{\text{sup}}. \qquad (32)$$

# E. Detailed Evaluation Metrics

## E.1. Gene Reconstruction Metrics

Let $g_{1:L}$ be the ground-truth serialized gene identity sequence and $\hat{g}_{1:L}$ the reconstructed sequence. Let $v_{1:L}$ and $\hat{v}_{1:L}$ be the corresponding expression values.

**L-Dist (Wasserstein-1 over ranks).** Let $\pi(g)$ be the rank (position index) of gene $g$ in the ground-truth list and $\hat{\pi}(g)$ the rank in the predicted list (restricted to the evaluated set). Define

$$\text{L-Dist}(g, \hat{g}) = \frac{1}{L} \sum_{k=1}^{L} |k - \hat{\pi}(g_k)|, \qquad (33)$$

which is equivalent to the 1-Wasserstein (earth mover) distance between the two uniform measures over ranks with ground cost $|i - j|$ under the induced permutation.

**BLEU for gene sequences.** Treat $g_{1:L}$ and $\hat{g}_{1:L}$ as token sequences. For $n$-grams ($n = 1, \ldots, N$), define clipped precision $p_n$ and the brevity penalty BP. BLEU-$N$ is

$$\text{BLEU-}N = \text{BP} \cdot \exp\left(\sum_{n=1}^{N} w_n \log p_n\right), \qquad (34)$$

with weights $w_n$ (typically $w_n = \frac{1}{N}$).

**Spearman correlation for values.** Compute Spearman's rank correlation between $\hat{v}$ and $v$ across evaluated genes:

$$\rho_{\text{sp}} = 1 - \frac{6 \sum_{j=1}^{m} d_j^2}{m(m^2 - 1)}, \qquad (35)$$

where $m$ is the number of evaluated genes and $d_j$ is the difference between ranks of $\hat{v}_j$ and $v_j$.

## E.2. Integration Metrics

Let $z_i \in \mathbb{R}^{d_z}$ be the learned cell embedding for cell $i$, with batch label $b_i$ and biological label (cell type) $c_i$. Let $\text{ASW}(\cdot)$ denote the average silhouette width computed from pairwise distances in embedding space.

**Avg-Batch.** We quantify batch mixing via

$$\text{Avg-Batch} = 1 - \text{ASW}_{\text{batch}}, \qquad (36)$$

where $\text{ASW}_{\text{batch}} = \text{ASW}(z_i, b_i)$ (lower silhouette for batch labels indicates better mixing).

**Avg-Bio.** We quantify biological conservation by combining clustering agreement and within-type compactness. Let $\widehat{c}_i$ be cluster assignments obtained from $z_i$. Define normalized mutual information $\text{NMI}(\widehat{c}, c)$, adjusted Rand index $\text{ARI}(\widehat{c}, c)$, and biological silhouette $\text{ASW}_{\text{bio}} = \text{ASW}(z_i, c_i)$. Then

$$\text{Avg-Bio} = \frac{1}{3}\left(\text{NMI} + \text{ARI} + \text{ASW}_{\text{bio}}\right). \qquad (37)$$

**Centered DE Pearson.** Standard DE Pearson can be influenced by shared global control-to-perturbation shifts. To evaluate perturbation-specific residual structure, we compute centered DE Pearson. Let $\Delta_{p,j}$ and $\widehat{\Delta}_{p,j}$ denote the observed and predicted differential expression for perturbation $p$ and gene $j$. Let $\bar{\Delta}_j$ and $\bar{\widehat{\Delta}}_j$ denote the average observed and predicted DE vectors across perturbations. The centered vectors are

$$\Delta_{p,j}^c = \Delta_{p,j} - \bar{\Delta}_j, \qquad \widehat{\Delta}_{p,j}^c = \widehat{\Delta}_{p,j} - \bar{\widehat{\Delta}}_j. \qquad (38)$$

C-DEP is the Pearson correlation between $\Delta_p^c$ and $\widehat{\Delta}_p^c$ over the evaluated DE genes, averaged across perturbations.

*Table 7.* **Downstream adaptation and optimization protocols.** FT denotes full fine-tuning annotation, ZS denotes zero-shot annotation with a frozen backbone, LR denotes learning rate, BS denotes batch size, and Ep. denotes the maximum number of fine-tuning epochs.

| Protocol | Rank-value reconstruction | Multi-batch integration | Cell type annotation (FT) | Cell type annotation (ZS) | Perturbation prediction |
|---|---|---|---|---|---|
| **Updated modules** | None; backbone frozen | Full encoder | Last 4 layers; first 8 frozen | MLP head only; backbone frozen | Full encoder |
| **Task head** | Identity and value denoising heads | GRL batch discriminator; projection MLP | MLP classifier | MLP classifier | `[PERT]` token; MLP regressor |
| **Loss** | $\mathcal{L}_{\mathrm{id}} + \lambda\mathcal{L}_{\mathrm{val}}$ | Reconstruction + batch adversarial + SupCon | Cross-entropy | Cross-entropy | Weighted MSE + DE-aware ranking term |
| **Optim.** | N/A | LR $1\times10^{-4}$; BS 256; Ep. 50 | LR $5\times10^{-5}$; BS 256; Ep. 30 | LR $5\times10^{-5}$; BS 256; Ep. 80 | LR $1\times10^{-4}$; BS 128; Ep. 40 |
| **Selection** | Dataset-specific evaluation | Validation Avg-bio / ASW-bio | Validation Macro-F1 | Validation Macro-F1 | Validation DE-AUPRC |

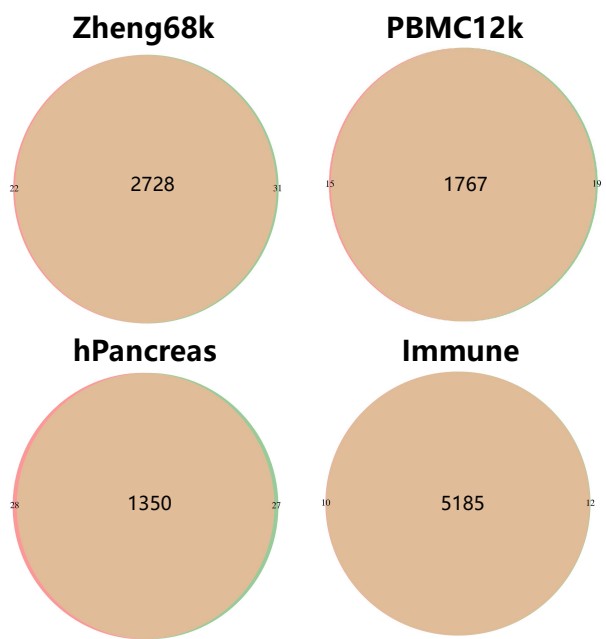

*Figure 8.* **Gene Space Reconstruction Overlap.** Venn diagrams illustrating the intersection of effectively reconstructed gene sets across Zheng68k, PBMC12k, hPancreas, and Immune datasets. The high intersection numbers demonstrate the model's ability to capture core transcriptomic features consistently across varying biological contexts.

**DE-AUPRC.** To evaluate recovery of differentially expressed genes, we treat experimentally identified DE genes as positives and rank genes by the predicted absolute perturbation effect $|\widehat{\Delta}_{p,j}|$. The area under the precision-recall curve is computed for each perturbation and then averaged across perturbations.

**LFCSpear.** LFCSpear measures the Spearman rank correlation between predicted and observed log-fold changes:

$$\mathrm{LFCSpear} = \rho_{\mathrm{sp}}\left(\widehat{\Delta}_{p,1:G}, \Delta_{p,1:G}\right). \qquad (39)$$

**HitRate@20.** HitRate@20 measures the fraction of experimentally identified top-20 DE genes recovered by the model-predicted top-20 genes:

$$\mathrm{HR@20} = \frac{|\mathrm{Top20}(\widehat{\Delta}_p) \cap \mathrm{Top20}(\Delta_p)|}{20}. \qquad (40)$$

## F. Additional Experimental Results

**Note on Inference Sampling:** Due to the high computational cost associated with multi-step diffusion sampling on large-scale benchmarks, all experimental results reported in this section are derived from single-step generation (i.e., direct prediction from the masked state), unless explicitly stated otherwise.

In this appendix, we provide extended qualitative and quantitative analyses to support the findings presented in the main text. We organize these results by the downstream tasks: Gene Reconstruction, Multi-batch Integration, Cell Type Annotation, Perturbation Prediction, and Gene Regulatory Network Inference.

### F.1. Extended Gene Reconstruction Analysis

To further validate the generative capacity of scDiVa, we analyzed the intersection of reconstructed features across diverse datasets. Figure 8 displays Venn diagrams illustrating the overlap of highly variable genes effectively reconstructed by the model across the Zheng68k, PBMC12k, hPancreas, and Immune datasets.

The substantial overlap indicates that scDiVa captures universal transcriptomic dependencies that are invariant across

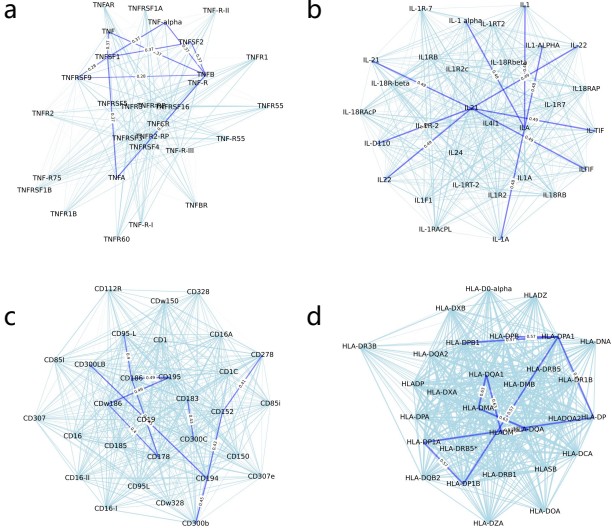

*Figure 9.* **Extended attention-derived gene association networks.** Network graphs constructed from scDiVa attention statistics for specific gene families: (a) TNF superfamily, (b) interleukins (IL-1 family), (c) CD surface markers, and (d) HLA complex. Thicker edges indicate stronger attention-derived association weights. These visualizations are intended as regulatory hypotheses rather than causal network reconstructions.

tissues. Specifically, the model maintains high fidelity not only for dataset-specific markers but also for shared housekeeping modules. This suggests that the Rank-Value reconstruction objective successfully prevents mode collapse and ensures that the generated expression profiles respect the underlying biological manifold of distinct datasets.

### F.2. Extended Multi-batch Integration

While the main text focuses on the Immune dataset, we evaluated scDiVa's integration performance on four additional challenging scenarios: Perirhinal Cortex (brain tissue), PBMC12k (blood), BMMC (bone marrow), and COVID-19 (pathological lung tissue).

As shown in Figure 10, scDiVa achieves a robust balance between batch mixing and biological conservation across all scenarios.

For the **Perirhinal Cortex** and **PBMC12k** datasets (Panels a, b), the UMAP projections colored by 'Batch' show thorough intermingling of samples (red and blue dots), indicating effective removal of technical artifacts. Conversely, the 'Cell type' plots reveal distinct, compact clusters, preserving the separation between neuronal subtypes and immune populations.

In the **BMMC** dataset (Panel c), which contains complex developmental trajectories, scDiVa preserves the continuum of differentiation (e.g., from HSCs to erythroid progenitors) while merging samples from different donors. This is evidenced by the unified trajectory in the batch plot and

gradient separation in the cell type plot.

Finally, for **COVID-19** (Panel d), the dataset introduces severe pathological heterogeneity. Despite strong disease-induced shifts, scDiVa successfully aligns the control and disease batches without erasing the disease-specific cell states, such as activated macrophages and T cells.

### F.3. Detailed Cell Type Annotation

To provide granular insight into the classification performance, we visualize the confusion matrices, predicted embeddings, and hierarchical heatmaps for representative fine-tuning tasks. In addition to the circular summary plot in Figure 4a, Table 8 reports the complete fine-tuning annotation results, including both the original baselines and additional recent foundation-model baselines under matched evaluation protocols.

We use the term "unseen" in a task-specific manner. For cell type annotation, the split is cross-source or cross-domain rather than an IID random split. hPancreas trains on Braon and Muraro and tests on Xin, Segerstolpe, and Lawlor. MS trains on healthy controls and tests on multiple sclerosis samples. Myeloid trains on UCEC, PAAD, THCA, LYM, cDC2, and KIDNEY and tests on MYE, OV-FTC, and ESCA. These settings evaluate cross-batch, cross-condition, or unseen-context transfer. For perturbation prediction, the split is by perturbation identity and evaluates unseen perturbation generalization, but it does not establish cross-cell-line generalization.

Figure 7 presents these results across four benchmarks: hPancreas, MS, Myeloid, and Myeloid_b. The comparison between "Annotated" ground truth and "Predicted" UMAPs in panels a, d, g, and j confirms that the model-learned latent space aligns closely with human-curated labels.

The **Confusion Matrices** in panels b, e, h, and k exhibit strong diagonal dominance, indicating high classification accuracy. Off-diagonal errors are mainly concentrated among biologically related subtypes, reflecting subtle transcriptomic differences rather than systematic model failure.

Furthermore, the **Hierarchical Heatmaps** in panels c, f, i, and l illustrate the distinct expression signatures learned for each predicted class. Their block-structured patterns indicate that scDiVa extracts discriminative features that are consistent within cell types and separable across classes.

### F.4. Perturbation Prediction Hit Rate

Beyond the correlation and MSE metrics reported in the main text, predicting the exact set of differentially expressed (DE) genes is critical for experimental prioritization. Figure 11 illustrates the Hit Rate at top-$k$ for the Norman dataset.

The Hit Rate metric quantifies the proportion of the true

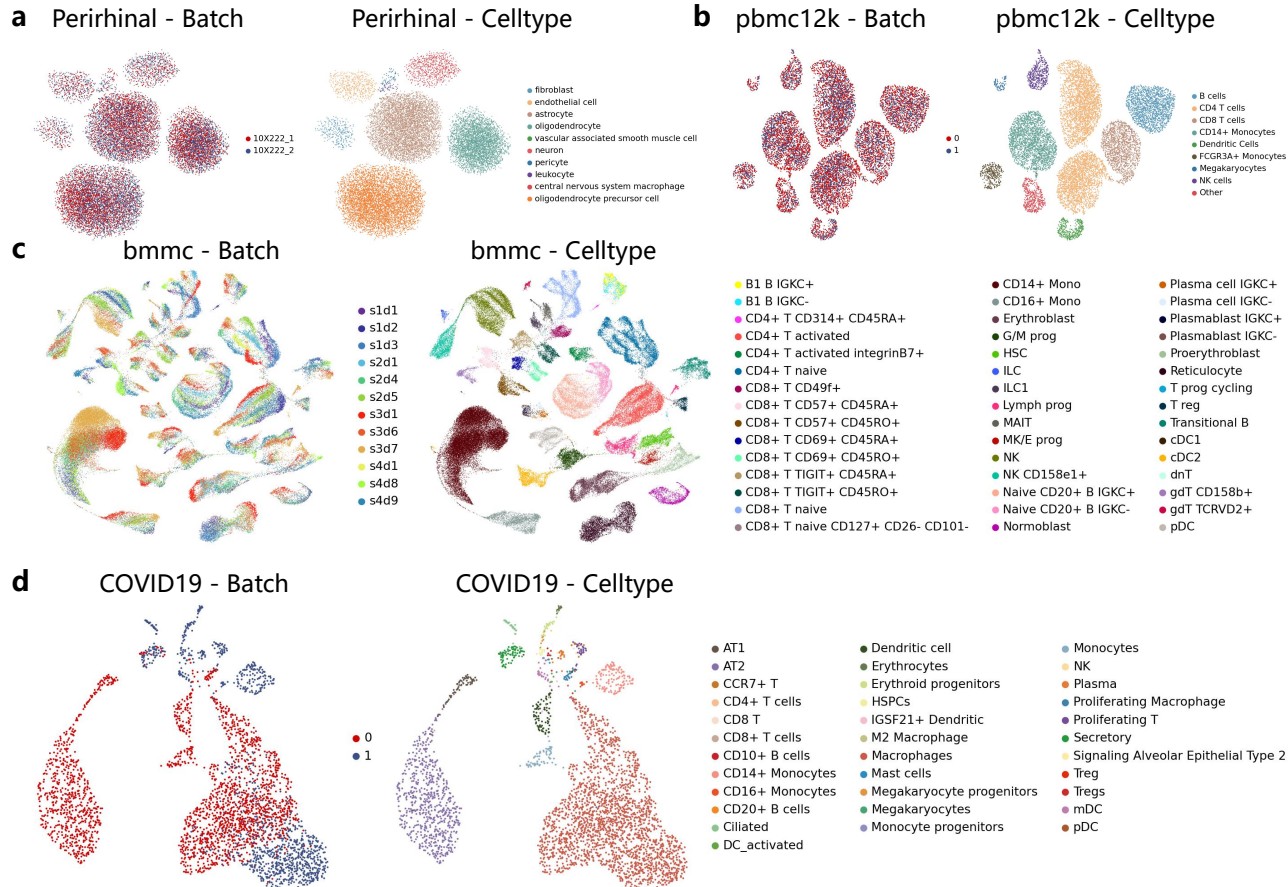

*Figure 10.* **Extended Multi-batch Integration Visualizations.** UMAP projections for (a) Perirhinal Cortex, (b) PBMC12k, (c) BMMC, and (d) COVID-19 datasets. Left columns display cells colored by batch ID, demonstrating effective mixing and removal of technical noise. Right columns display cells colored by cell type, confirming the preservation of biological identity and structural heterogeneity.

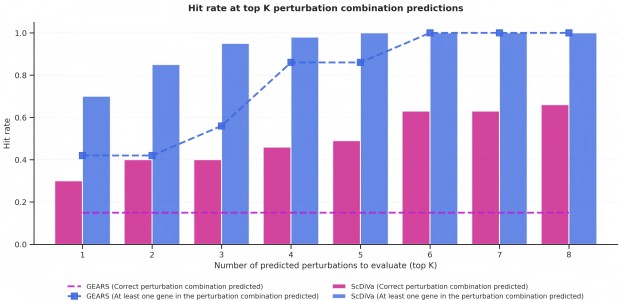

*Figure 11.* **Hit Rate at Top-$k$ for Perturbation Prediction.** Evaluation on the Norman dataset showing the proportion of true differentially expressed genes successfully retrieved by the model within the top $k$ predictions. scDiVa demonstrates superior ranking capability compared to baselines.

top-$k$ most responsive genes that are correctly identified by the model. scDiVa consistently outperforms baseline methods across varying $k$ values. This suggests that even when the exact magnitude of expression change is challenging to predict, our model correctly ranks the genes most affected by the perturbation, providing valuable candidates

for downstream wet-lab validation.

### F.5. Inferred Gene Regulatory Networks

The attention statistics within scDiVa can be used to construct gene association networks for regulatory hypothesis generation. In the main text, we analyzed the SPI1-centered neighborhood as a focused example. Here, we extend this analysis to several biologically relevant gene families to examine whether the learned association patterns are consistent with known pathway-level organization.

Table 9 provides the statistical context for the attention-derived GRN analysis. We report both the global network construction statistics and the SPI1-specific enrichment test. The random-overlap baseline is computed under the same background gene set, which helps assess whether the observed SPI1 target overlap can be explained by chance rather than by network density alone.

Figure 9 displays the inferred gene regulatory networks for four distinct biological modules:

*Table 8.* **Fine-tuning annotation benchmark under matched protocols.** We report Accuracy and Macro-F1 on four cross-batch or cross-domain annotation datasets. The table complements Figure 4a by providing the complete numerical results, including the original baselines and additional recent foundation-model baselines.

| Metric | Dataset | GeneFormer | scGPT | scFoundation | GeneMamba | CellFM | UCE | GeneCompass | scDiVa |
|--------|---------|------------|-------|--------------|-----------|--------|-----|-------------|--------|
| **Accuracy** | hPancreas | 96.7 | 97.1 | 96.0 | 97.1 | 98.1 | 97.8 | 98.0 | **98.6** |
| | MS | 76.5 | 84.7 | 77.6 | 68.3 | **85.6** | 81.8 | 82.5 | 84.4 |
| | Myeloid | 64.5 | 63.4 | 64.5 | 66.1 | 67.4 | 64.8 | 67.6 | **68.3** |
| | Myeloid_b | 95.4 | 94.2 | 95.7 | 96.0 | 95.8 | 94.6 | 95.0 | **96.0** |
| **Macro-F1** | hPancreas | 74.5 | 76.3 | 71.1 | 77.1 | 77.8 | **80.1** | 78.0 | 79.2 |
| | MS | 62.2 | 66.3 | 68.1 | 53.4 | 70.5 | 70.8 | 70.4 | **72.7** |
| | Myeloid | 36.0 | 35.6 | 36.5 | 36.5 | 37.6 | 30.8 | 35.5 | **40.2** |
| | Myeloid_b | 93.8 | 91.3 | **95.7** | 92.4 | 94.8 | 93.4 | 94.1 | 95.6 |

*Table 9.* **Statistical context for attention-derived GRN analysis.**

| Quantity | Value |
|----------|-------|
| Global GRN genes | 12,303 |
| Candidate undirected edges | 75,675,753 |
| Threshold | Top 0.01% |
| Retained edges | 7,568 |
| SPI1 background genes | 5,060 |
| SPI1 database targets in background | 120 |
| Top-$k$ neighbors | 20 |
| Observed overlap | 4 |
| Odds ratio | 10.6 |
| Fisher exact test $p$ | $1.1 \times 10^{-3}$ |
| Random overlap mean $\pm$ std | $0.47 \pm 0.68$ |
| Empirical $p$ | 0.006 |

**(a) TNF Superfamily:** The model recovers ligand-receptor pairs, specifically the interaction between TN-FSF13B and TNFRSF13B.

**(b) Interleukin Family (IL-1):** The graph exhibits high connectivity within the IL-1 family, consistent with the coordinated expression patterns of pro-inflammatory cytokines.

**(c) CD Markers:** We observe distinct co-expression clusters corresponding to cell-surface markers.

**(d) HLA Complex:** The attention mechanism identifies the dense correlation structure among MHC Class II genes (HLA-DR, -DQ, -DP).

Edge weights represent attention-derived association scores. These patterns should be interpreted as biologically plausible hypotheses rather than validated regulatory or protein-protein interactions.

*Table 10.* **LAT ablation under high masking ratios.**

| Mask ratio | w/ LAT Acc. ↑ | w/o LAT Acc. ↑ |
|------------|---------------|----------------|
| 70% | 0.91 | 0.89 |
| 80% | 0.86 | 0.81 |
| 90% | 0.78 | 0.66 |
| 95% | 0.64 | 0.47 |

*Table 11.* **Ordering sensitivity and RoPE ablation.** ZS F1 denotes zero-shot Macro-F1. Random permutation results are averaged over five test-time permutations.

| Setting | ZS F1 ↑ | Imm. $\rho$ ↑ | Norm. LFCSp ↑ |
|---------|---------|---------------|---------------|
| Full + RoPE | 0.840 | 0.970 | 0.433 |
| No RoPE | 0.830 | 0.960 | 0.430 |
| Random perm. | 0.839±0.002 | 0.960±<0.001 | 0.450±0.004 |

# G. Component Ablations and Robustness Analyses

## G.1. Latent Anchor Token Ablation

The latent anchor token `[LAT]` is designed to stabilize cell-level identity under high masking ratios. Table 10 shows that the benefit of `[LAT]` becomes more pronounced as the masking ratio increases.

## G.2. Ordering Sensitivity and RoPE Ablation

Although scDiVa reduces autoregressive ordering bias by avoiding causal masking, it is not strictly permutation invariant because deterministic serialization and RoPE are used. We therefore evaluate sensitivity to RoPE removal and test-time random permutations under fixed gene sets.

Table 11 reports that removing RoPE or applying random test-time permutations produces only small changes across zero-shot annotation, reconstruction, and perturbation-oriented metrics.

*Table 12.* **Serialization strategy ablation under a fixed token budget.** Recon $\rho$ denotes reconstruction Spearman correlation, and ZS F1 denotes zero-shot Macro-F1.

| Strategy | Recon $\rho$ ↑ | ZS F1 ↑ |
|---|---|---|
| Entropy-normalized | 0.97 | 0.84 |
| Expression sorting | 0.94 | 0.82 |
| Random subset | 0.90 | 0.79 |

*Table 13.* **Depth-robust corruption ablation under depth down-sampling.** Macro-F1 is evaluated at different test-depth levels.

| Train corruption | 0.25× ↑ | 0.5× ↑ | 1.0× ↑ |
|---|---|---|---|
| Mask-only | 0.72 | 0.80 | 0.84 |
| Global scaling + mask | **0.77** | **0.82** | 0.84 |

### G.3. Entropy-Normalized Serialization Ablation

We frame entropy-normalized serialization as an information-allocation strategy under a fixed token budget, rather than as a complete solution to sparsity. Under the same sequence length, entropy-normalized ordering outperforms expression-only sorting and random gene subsets.

Table 12 shows that entropy-normalized serialization improves both reconstruction correlation and zero-shot Macro-F1 under the same token budget.

### G.4. Depth-Robust Corruption Ablation

Gene-level masking is not an exact physical model of sequencing-depth reduction. A true reduction in sequencing depth globally thins molecular counts, whereas masked discrete diffusion removes information at the gene-token level. These two mechanisms are related through dropout-like non-detection, but they are not equivalent. To empirically probe this gap, we compare mask-only corruption with a mixed corruption strategy that combines global scaling or downsampling with gene-token masking.

Table 13 shows that mixed corruption improves Macro-F1 under stronger depth reduction, while matching mask-only training at the full-depth setting. This supports the use of depth-robust corruption as a sparse-observation robustness strategy, not as a mechanistically exact model of library-size variation, capture efficiency, or amplification bias.

