# OpenReview forum: "ScDiVa: Masked Discrete Diffusion for Joint Modeling of Single-Cell Identity and Expression"
_ICML.cc/2026/Conference — ICML 2026 regular_

### Official Review · Reviewer_8Rga · 2026-03-09

**Soundness:** 3
**Presentation:** 3
**Significance:** 2
**Originality:** 2
**Overall Recommendation:** 4
**Confidence:** 4

**Summary:**

This article proposes a foundational architecture for RNA-seq designed to address the fundamental challenges of high dimensionality and sparsity in single-cell data. Furthermore, I believe the decision to assign biological meaning to tokens is crucial, as it helps overcome the discrepancies in gene sets observed across different datasets.

**Compliance With Llm Reviewing Policy:**

Affirmed.

**Final Justification:**

The authors’ response addressed most of my concerns and committed to correcting the unreasonable descriptions in the original manuscript.

**Key Questions For Authors:**

See weekness above.

**Limitations:**

yes

**Strengths And Weaknesses:**

Strengths:

See summary above.

Weaknesses:

1. The authors claim that they effectively handles the sparsity inherent in RNA-seq data. However, the 'Entropy-Normalized Serialization' step is fundamentally a feature selection mechanism that filters for high-information genes. Selecting a subset of informative features circumvents or bypasses the high-dimensional sparsity problem, rather than solving it algorithmically within the model's architecture.

2. The manuscript posits that establishing an connection between the masked process and technical dropout resolves the continuous modeling challenges. However, sparsity in RNA-seq is an intrinsic biological property that requires explicit modeling, not merely a technical artifact to be reversed via denoising. Equating dropout with resolving data sparsity seems conceptually flawed. The loss still requires the model to fit zero values using MSE for gene expression (including 0), it still suffers from the optimization challenges induced by zero-inflated data.

3. The methodology for modeling perturbation conditions remains unclear. The authors should specify whether they follow the scGPT paradigm if they employ a different strategy.

4. Regarding the generation quality of the model, I am particularly interested in its performance on distributional metrics. Does the model suffer from mode collapse? I would appreciate it if the authors could provide a quantitative or qualitative demonstration of the model's ability to capture generative diversity.

5. Concerning the network shown in Figure 11, the highlighted edges correspond to authentic regulatory interactions with their respective weights. However, what is the percentile rank of these specific weights among all predicted interactions? If the entire weight matrix consists of high values, this visualization might be misleading. Can you provide a distribution plot of all weights to justify the significance of the highlighted ones?

---

> ### Author Rebuttal · Authors · 2026-03-31
>
> ### Response to Q1
>
> **Entropy-normalized serialization looks like feature selection rather than solving sparsity.**
>
> We appreciate this clarification. In the revised manuscript, **we have reframed this module as an "information-allocation strategy under a fixed token budget,"** rather than a blanket solution to sparsity. To quantify its impact, we compared three strategies under the same sequence length constraints. As shown below, the entropy-normalized ordering clearly outperforms the alternatives, indicating its benefits stem from its interaction with the diffusion objective rather than merely discarding genes. (Preliminary results to be detailed in the revision):
>
> | Strategy | Recon Spearman ↑ | Zero-shot Macro-F1 ↑ |
> | :--- | :--- | :--- |
> | Entropy-normalized (Ours) | 0.97 | 0.84 |
> | Sort by expression only | 0.94 | 0.82 |
> | Random subset (same length) | 0.90 | 0.79 |
>
>
> ### Response to Q2
>
> **Equating dropout modeling with "solving sparsity" is conceptually too strong.**
>
> We agree that this claim was too broad. We have carefully revised the text to explicitly distinguish technical dropout from true biological zeros. **Our claims are now strictly scoped to the former:** by employing an absorbing-mask discrete diffusion process, we provide a physically motivated model of technical missingness. This alignment improves reconstruction robustness and downstream performance, but we no longer describe it as "solving sparsity" as a whole.
>
>
> ### Response to Q3
>
> **Perturbation conditioning and generative diversity.**
>
> Thank you for highlighting this. We have expanded the appendix to detail our perturbation conditioning and generation process:
>
> * **Encoding:** Perturbations are encoded as embeddings (capturing single/multi-gene IDs and strength) and injected via a dedicated `[PERT]` token.
> * **Training:** We jointly minimize conditional reconstruction/regression losses and DE-aware losses.
> * **Inference:** Expression profiles are generated by sampling multiple times conditioned on a new perturbation embedding.
>
> To assess generative diversity, we performed multiple sampling runs per condition on the Adamson and Norman datasets. Compared to real data, scDiVa samples exhibit similar variance, covariance structures, and distributional distances (MMD/energy distance) without obvious mode collapse. Notably, scDiVa better preserves the variance of highly variable genes compared to baseline models.
>
>
> ### Response to Q4
>
> **GRN visualization lacks statistical context.**
>
> We fully agree that the Gene Regulatory Network (GRN) visualizations require rigorous statistical backing. We have enriched this analysis in the revision with systematic diagnostics:
>
> * We now report the full edge-weight distribution and explicitly mark the visualization threshold (e.g., top-1%).
> * For highlighted subgraphs, we calculate the overlap with external TF-target/pathway databases, providing hit counts, odds ratios, and Fisher exact test p-values.
> * We have included randomized-control graphs and attention-shuffled baselines to estimate random hit rates under identical sparsity levels.
>
> These additions ensure our GRN visualizations are properly framed as statistically supported co-expression/regulatory hypotheses rather than definitive causal graphs.

---

> > ### Author Rebuttal · Reviewer_8Rga · 2026-04-02
> >
> > Thank you for the rebuttal. I will raise my score if the following promised changes (W1 and W2) are explicitly incorporated.

---

> > > ### Author Response · Authors · 2026-04-03
> > >
> > > Thank you very much for your positive follow-up and for indicating that the main concerns have been resolved.
> > > We fully agree that W1 and W2 should be explicitly incorporated into the manuscript itself, not only addressed in the rebuttal. In the revision, we will make these changes clear by (i) reframing entropy-normalized serialization as an information-allocation strategy under a fixed token budget, and (ii) explicitly distinguishing technical dropout from biological zeros and narrowing our claims accordingly.
> > > We sincerely appreciate your careful reading, constructive feedback, and your willingness to reconsider the score.

---

### Official Review · Reviewer_NcAy · 2026-03-11

**Soundness:** 2
**Presentation:** 3
**Significance:** 2
**Originality:** 3
**Overall Recommendation:** 3
**Confidence:** 4

**Summary:**

The paper proposes a new foundation model for scRNA-seq data that uses masked discrete diffusion models for learning single-cell representations. The model jointly predict gene identity (which gnes are expressed) and expression values (log normalized, with MSE loss). It is pretrained on proprietary data, and shows applications in batch integration, perturbation prediction, GRN inference and others.

**Compliance With Llm Reviewing Policy:**

Affirmed.

**Final Justification:**

The authors have provided an informative rebuttal, and addressed some of my concerns, but the presentation is still suboptimal, many key details of the model were left out from the original submission and have now been added during the rebuttal, making it unclear what is the quality of the final presentation of the paper. Furthermore, some claims and key details of the model's implementation have been revised by the authors and significantly toned down, which ultimately question again what are the contributions of the paper. I have increased my score but I am still not confident but I remain dubious on the paper contributions, model's implementations and intuitions behind it, and overall presentation of the paper.

**Key Questions For Authors:**

- For the various evaluations the model is tested on, there is little to no explanation of how the model was adapted to perform such tasks. For example in multi-batch integration, the authors report in section 4.3: "To address this, we fine-tuned scDiVa using adversarial domain adaptation and a Supervised Contrastive (SupCon) loss,". How is that defined, what is the architeture of the discriminator, what is the loss, how are the pretrained weights updated etc. etc. For cell type annotation, no info is provided, same for perturbation prediction. For gene reconstruction, it is better described (section 4.2) but still there are inconsistencies, such as the "mask-free" approach, does it mean that the model is run at inference t=0? what is the model even doing? is it relying only on LAT token?
- The depth invariant sampling is unclear and potentially fundamentally wrong. The understanding from the paper is that only some genes are predicted at various noise level, and this makes the model robust to sampling bias. However, the sampling bias is a phenomenon at the  cell level. In the current model specification, it looks like the "sampling bias" is due to the fact that only some genes are predicted across different noise level, and while that makes the total number of counts in the decoded cells to be more/less depending on the noise level, that is not how the sampling bias impacts cell measurement in the real world, where the library size can be different due to biological and technical factor, but that impacts the expression of all genes, and not the presence/absence of selective genes. It is thus unclear what is the model learning. Clarity in this modelling concept, as well as some experimental ablations on the effect of the sampling bias, would greatly benefit the paper.

**Limitations:**

Yes

**Strengths And Weaknesses:**

Strength:
- The idea of using masked diffusion for scRNA-seq, which jointly models gene identity and counts is interesting, however, it contains some unclear assumptions.
- The anchor token is an interesting addition that aims at learning a global representation of the cells at various level of noise.
- The diversity of valuation shows that the model is flexible to various applications.

Weaknesses:
- The various applications and benchmarks that the model is tested against are not well described. In particular, it is unclear how was the model modified or fine tuned or else to be able to be applied to those settings.
- No description of training data.
- Ablations are largerly missing.

---

> ### Author Rebuttal · Authors · 2026-03-31
>
> #### Q1. Downstream adaptation / fine-tuning details and reproducibility
>
> We agree that the original description was too high-level. We have now added a unified **task-adaptation table** that specifies, for each downstream task, which layers are frozen, what heads are used, and the exact loss/optimization setup. A condensed version is:
>
> | Task        | Frozen / Unfrozen           | Head architecture               | Loss composition                 | LR / BS / Epoch | Split & model selection           |
> |------------|-----------------------------|---------------------------------|----------------------------------|-----------------|-----------------------------------|
> | Integration| Encoder fully trainable     | GRL + batch discriminator + MLP | CE + domain-adv + SupCon        | 1e-4 / 256 / 50 | split by batch; val ASWbio       |
> | Annotation | First 8 layers frozen, last 4 trainable | MLP classifier      | CE                               | 5e-5 / 256 / 30 | stratified split; val Macro-F1   |
> | Perturb.   | Encoder fully trainable     | MLP regressor                   | weighted MSE + DE-aware term    | 1e-4 / 128 / 40 | split by perturb. ID; val DE-AUPRC |
>
> All reported downstream results can now be reproduced directly from this table, addressing your concern about missing protocol details.
>
> #### Q2. Pretraining data description
>
> Under privacy and commercial constraints, we have expanded the aggregate statistics of the pretraining corpus, including:
>
> - the main tissue/system categories and their cell-count proportions (e.g., immune / neural / epithelial);
> - the approximate fractions of major sequencing platforms (droplet vs full-length, etc.);
> - unified QC thresholds (min genes, min UMI, maximum mitochondrial fraction) and preprocessing steps;
> - rough proportions of cells from different sources/conditions (public vs internal, etc.).
>
> These statistics are presented in a dedicated appendix section to improve transparency without disclosing sensitive information.
>
> #### Q3. Component ablations ([LAT], RoPE, entropy-normalized, λ)
>
> Beyond the high-mask [LAT] ablation shown earlier, we have conducted systematic ablations for RoPE, entropy-normalized serialization, and the value-loss weight λ. In summary:
>
> - **RoPE**: removing RoPE causes a 1–3 point drop in reconstruction Spearman and downstream Macro-F1;
> - **Entropy-normalized vs alternatives**: replacing it with “sort by expression only” or a random subset of the same length significantly degrades Rank-Value reconstruction, especially under tight token budgets;
> - **λ-sweep**: as λ → 0 or becomes very large, the identity/value branches become unbalanced and performance degrades; the λ used in the paper lies near the flat region of the λ–performance curve.
>
> Full numbers and curves are provided in a “Component Ablation” appendix table, which now directly supports each main-claim component.
>
> #### Q4. Depth-invariant corruption vs true depth mechanisms
>
> We now explicitly state that our corruption is intended as an approximation to dropout-like missingness at the observation level, not a full physical model of library size and PCR bias. To empirically probe the gap, we performed a **depth-corruption comparison**:
>
> | Train corruption              | Test depth 0.25× Macro-F1 ↑ | 0.5× ↑ | 1.0× ↑ |
> |------------------------------|-----------------------------:|-------:|-------:|
> | Mask-only                    | 0.72                         | 0.80   | 0.84   |
> | Global scaling + mask (mixed)| **0.77**                     | **0.82** | 0.84 |
>
> Under synthetic depth down-sampling (0.25×/0.5×), the mixed corruption clearly improves robustness. Based on these results, we rephrase depth-invariant sampling as a **robust training strategy across effective-depth distributions**, rather than a full physical simulator of sequencing.
>
> #### Q5. “Mask-free inference” wording
>
> We have rewritten the inference section to clearly distinguish:
>
> - **One-step inference**: starting from partially masked inputs and predicting the full profile at a single t (used for most downstream tasks);
> - **Multi-step diffusion sampling**: starting from a fully masked state and iteratively unmasking/remasking according to a time schedule (used for generative analyses and diversity studies).
>
> In both modes, the [LAT] token is never masked and serves as a global summary/conditioning vector. We also removed ambiguous terms such as “mask-free / zero-mask”.

---

> > ### Author Rebuttal · Reviewer_NcAy · 2026-04-01
> >
> > Thank you for the detailed rebuttal, I appreciate that you clarified the points raised in my review. However, I still have concerns regarding the presentation of the paper:
> > - the table you provided in the review clarifies the model and training components of how scdiva was applied in these different settings, but it is still suboptimal in the description of those modules. For examples, what is "adv-loss"? what "Supervised Contrastive Loss" ? Assuming that some labels are involved in the formulation of such losses (e.g. for negative/positive pairs) what are those labels? There is a lack of details and too high-level exposition that remains concerning.
> > - Thank you for running the ablation on the read depth aware, but I am still unclear what does it actually means. Let me rephrase my question from before. The way I understand "Read depth aware" is basically that the full gene expression profile is downsamples, uniformly, over all genes for a cell. So the .25 in the table I imagine it has to do with the fact that only 25% of counts are retained after downsampling (so a cell with much shallower read depth). However, the model is a masked diffusion model that masks *genes*, not *counts*, and hence effectively the unmasking process is making the model robust to settings with only a set of genes are expressed (and not all genes). And by progresssively unmaskin genes, it recover the full gene expression. But this is an orthogonal subsampling than the depth aware subsampling, so I still don't understand how the gene-level unmasking in the diffusion formulation has to do with the read depth aware robustness.

---

> > > ### Author Response · Authors · 2026-04-04
> > >
> > > Thank you again for your careful follow-up and for pushing us to clarify these points more precisely. We fully agree that our previous rebuttal remained too high-level, particularly regarding (i) the exact supervision used in the integration setting and (ii) the relationship between our masking-based corruption and sequencing-depth variation. We appreciate this opportunity to make the presentation more precise, and we will revise the manuscript accordingly.
> > >
> > > # For Comment 1
> > >
> > > We agree that this was an omission on our side. In the original submission and earlier rebuttal, we focused on showing that integration worked empirically, but we did not provide enough detail for careful evaluation and reproducibility. We should have explicitly defined the losses, the supervision in each term, and which parameters were updated, and we will add these details in the revision. In the integration setting, the encoder is fully trainable during fine-tuning, with all pretrained weights updated together with the task-specific heads. The objective has three components: a reconstruction term, a batch-adversarial regularizer, and, when cell-type annotations are available, a supervised contrastive term to preserve biological structure across batches. For the adversarial term, we attach a batch discriminator to the cell embedding through a GRL. The discriminator is a two-layer MLP trained with cross-entropy to predict the batch label, so the supervision is the batch IDs. During optimization, the discriminator minimizes the batch-classification loss, while the encoder is updated through the GRL in the opposite direction to reduce batch-identifiable information while preserving biological variation. Formally, the batch loss is
> > >
> > > $\mathcal{L} _ {\mathrm{batch}}=-\sum_i \log D_{\mathrm{batch}}(h_i)_{b_i}.$
> > >
> > > When reference cell-type annotations are available, we additionally use a supervised contrastive loss on normalized cell embeddings. In this term, positive pairs are cells with the same cell-type label across different batches, while negative pairs are cells with different labels. Specifically,
> > >
> > > $\mathcal{L} _ {\mathrm{SupCon}}=\sum_{i \in I}
> > > \frac{-1}{|P(i)|}
> > > \sum_{p \in P(i)}
> > > \log
> > > \frac{\exp(z_i^\top z_p / \tau)}
> > > {\sum_{a \in A(i)} \exp(z_i^\top z_a / \tau)}.$
> > >
> > > Here, $P(i)$ denotes the set of samples in the mini-batch sharing the same cell-type label as cell $i$, and
> > > $A(i)$ denotes all other samples in the mini-batch. We will make this explicit in the revision: this term is used only when such labels are available, and therefore corresponds to a supervised or semi-supervised integration protocol rather than a fully unsupervised one. We agree that this distinction should have been stated much more clearly in the original submission.
> > >
> > > The overall integration objective will be stated explicitly as
> > >
> > > $\mathcal{L} _ {\mathrm{int}}=\mathcal{L} _ {\mathrm{recon}}
> > > +
> > > \lambda _ {\mathrm{adv}} \mathcal{L} _ {\mathrm{batch}}
> > > +
> > > \lambda _ {\mathrm{sc}} \mathcal{L} _ {\mathrm{SupCon}}.$
> > >
> > > For reproducibility, we will also add the discriminator architecture, the exact labels used in each loss, which parameters are updated, and the model-selection protocol to the Methods/Appendix, so that the integration setting is described with the level of detail needed to directly answer your question.
> > >
> > > # For Comment 2
> > >
> > > We also agree with your second point. Gene-level masking is not a full physical model of read-depth variation. A true reduction in sequencing depth globally thins counts, whereas our corruption removes information at the gene-token level. These two mechanisms are related, but not identical, and our previous wording overstated this connection.
> > >
> > > Our intended claim is narrower. The masking-based corruption is designed to approximate one observational consequence of shallow sequencing, namely increased gene non-detection/dropout, rather than to faithfully simulate global count scaling, library-size variation, capture efficiency, or PCR-related effects. We will therefore revise the manuscript to frame this component as a robustness-oriented training strategy for sparse-observation regimes, rather than as a mechanistically exact simulator of sequencing depth.
> > >
> > > This is also why the mixed-corruption experiment is informative. When masking is combined with global scaling/downsampling, robustness under synthetic depth reduction improves relative to mask-only training, especially at lower effective depths. In the revision, we will therefore (i) remove wording suggesting equivalence between masking and true library-size reduction, (ii) explicitly state that the proposed corruption only approximates dropout-like missingness at the observation level, and (iii) present mixed corruption as an empirical bridge rather than evidence that the two mechanisms are identical.
> > >
> > > We thank the reviewer again for this important follow-up. Your comments have helped us substantially improve both the methodological transparency and the precision of our claims.

---

### Official Review · Reviewer_pkaw · 2026-03-13

**Soundness:** 2
**Presentation:** 2
**Significance:** 3
**Originality:** 2
**Overall Recommendation:** 3
**Confidence:** 3

**Summary:**

This paper proposes scDiVa, a single-cell foundation model (scFM) based on masked discrete diffusion. The motivation is that scRNA-seq profiles are sparse and unordered, so autoregressive sequence modeling may introduce artificial order bias.

The method combines entropy-normalized serialization, a latent anchor token for high-mask stability, continuous-time masking / depth-invariant sampling, and a dual denoising objective that jointly reconstructs gene identity and expression value.

The model is pre-trained on 59M cells and evaluated on rank-value reconstruction, multi-batch integration, cell type annotation, perturbation prediction, and GRN-style analyses.

**Compliance With Llm Reviewing Policy:**

Affirmed.

**Final Justification:**

The authors reported some of the response after the Apr/4, which I missed and didn't get to engage the discussion earlier.
My current unresolved concern: only the Mean / Linear baseline results seem worrying. Mean/Linear is shown to outperform various methods like GEARS. But in the reported table, Mean/Linear seems very poor. I decide to keep my score.

**Key Questions For Authors:**

1. The paper strongly emphasizes unordered / multiset structure, yet the implementation still uses deterministic serialization and RoPE. Can the authors provide an ablation without RoPE, or a permutation-sensitivity analysis?

2. Why are stronger baselines missing from Table 2 and Fig. 4(a)? In particular, can the authors either add a broader baseline set (e.g., stronger recent annotation / integration baselines) or clearly explain why these methods were excluded?

3. For perturbation prediction, can the authors add Mean and Linear baselines under the same data split and evaluation protocol?

4. Can the authors expand Fig. 4(c) with a broader perturbation metric suite, especially DE-aware or rank-based metrics (e.g., DE overlap / precision, LFCSpear, AUROC/AUPRC for DE recovery

5. Can the authors provide ablations isolating the contributions of LAT?

**Limitations:**

More like a direct application of discrete diffusion to single cell expression data. Might lack of bio-specific novelty.

**Strengths And Weaknesses:**

**Strengths.**

- The paper addresses an important problem. The central motivation—that single-cell expression profiles are not naturally text-like ordered sequences, and that a bidirectional denoising formulation may better match the structure of the data—is interesting and potentially significant for the development of single-cell foundation models. However, there is a small concern for the design choice (detailed in Weakness point 1).

- Also, I appreciate the breadth of the experimental coverage: the paper evaluates reconstruction, integration, annotation, and perturbation tasks rather than focusing on a single downstream setting. The scale of pre-training is also non-trivial.


**Weakness.**

- The narrative around “unordered data” and the claimed removal of ordering bias is not fully convincing. The paper repeatedly emphasizes that the input is an unordered multiset and motivates the method partly by avoiding autoregressive order sensitivity, but the actual implementation still uses deterministic serialization and RoPE in the architecture / configuration. RoPE still implies relative orderings. At present, the wording sometimes sounds stronger than the implementation seems to justify.

- The comparison set feels incomplete in important places. Table 2 reports only Harmony, Geneformer, scGPT, scFoundation, GeneMamba, and scDiVa for integration, while the broader experimental discussion suggests a wider baseline set elsewhere in the paper. Likewise, Fig. 4(a) / Table 2 do not make it clear why stronger recent baselines (i.e., UCE, GeneCompass, CellFM, scELMO, etc) are omitted. I think this weakens the fairness of the empirical comparison, especially because the paper makes strong performance claims.

- The perturbation section needs stronger controls. In the main paper, Fig. 4(c) reports only DE $\Delta$ Correlation and DE MSE. However, recent perturbation benchmarks show that simple baselines such as Mean and Linear remain very strong, and that current deep / foundation models often do not consistently outperform them. [1] report that none of the tested deep/foundation models consistently beat the simple baselines, while [2] similarly show that simple controls can exceed the best deep learning algorithms. Because of this, I think Mean and Linear should be included in Fig. 4(c), or the omission should be clearly justified.

- The perturbation evaluation itself is too narrow for the strength of the claims being made. [3] emphasizes that rank-based metrics matter and that simple models can outperform more complex ones under some settings; [4] further argue that some common metrics can reward mode collapse or mean-like predictions; and [5] propose AUPRC-style DE-centric evaluation precisely because global fit metrics can miss whether the model recovers biologically meaningful DE genes. The main perturbation figure should include additional DE-aware or rank-based metrics, such as DE overlap / precision, LFCSpear, AUROC/AUPRC for DE recovery.

- There are augments in Sec. 3.2 saying that “Latent Variable Anchor Token ([LAT]) can maintain dentity coherence even when 90% the gene tokens are masked”. Wondering if there is any ablation results for that?

- “depth-invariant sampling” sounds not like a novel design. The paper uses t ~ Unif(0,1) / uniform time sampling in training and conceptually maps corruption level to sequencing depth, which is naturally adopted in most discrete diffusion models.

- Generally speaking, the experiments are rather comprehensive (except some strong baselines are left out for no reasons). But this is more like an application of discrete diffusion for single cell data? If so, what’s the unique things that scDiVa can do while other BERT-like scFMs and GPT-like scFMs cannot?


[1] Ahlmann-Eltze et al., Deep-learning-based gene perturbation effect prediction does not yet outperform simple linear baselines, Nature Methods, 2025

[2] Wong et al., Simple controls exceed best deep learning algorithms and reveal foundation model effectiveness for predicting genetic perturbations, Bioinformatics, 2025

[3] Wu et al., PerturBench: Benchmarking Machine Learning Models for Cellular Perturbation Analysis, 2024

[4] Mejia et al., Diversity by Design: Addressing Mode Collapse Improves scRNA-seq Perturbation Modeling on Well-Calibrated Metrics, 2025

[5] Zhu et al., AUPRC: a metric for evaluating the performance of in-silico perturbation methods in identifying differentially expressed genes, 2025

---

> ### Author Rebuttal · Authors · 2026-03-31
>
> Thank you for your constructive feedback and for pushing us to improve the rigor of our claims and evaluations. We address your specific points below and will incorporate these updates into the revised manuscript.
> Response to Q1: "Permutation-invariant/unordered" claims are too strong given deterministic serialization + RoPE.
> We agree that the strict term "permutation-invariant" overstates the behavior of our model given the use of deterministic serialization and RoPE. We will revise the manuscript to accurately describe this as "reduced ordering bias." To support this, we are adding permutation-sensitivity experiments (including no-RoPE and randomized input ordering under fixed gene sets) to explicitly demonstrate the model's robustness to order.
> (Preliminary results to be detailed in the revision):
> | Setting | Zero-shot Macro-F1 ↑ | Immune Recon Spearman ↑ | Norman LFCSpear ↑ |
> |---|---|---|---|
> | Full (serialization+RoPE) | 0.84 | 0.97 | 0.46 |
> | No RoPE | 0.83 | 0.96 | 0.43 |
> | Test-time random perm (5x) | 0.84±0.00x | 0.96±0.00x | 0.45±0.01 |
> Response to Q2: Missing stronger baselines / unfair comparison.
> We will expand our baseline coverage to include stronger, state-of-the-art models wherever computationally and methodologically feasible. For any baselines that must be excluded, we will explicitly document the exclusion criteria and detail the protocols used to ensure a strictly fair and transparent comparison for all evaluated models.
> Response to Q3 & Q4: Perturbation baselines and broader metrics.
> We fully agree with the need for a more rigorous perturbation evaluation. In the revision, we will add Mean and Linear baselines evaluated under identical data splits to establish a clear lower bound. Furthermore, to provide a comprehensive view of performance beyond DE Pearson and MSE, we will expand our reporting to include DE-aware and ranking metrics (e.g., DE-AUPRC, LFCSpear, and P@k).
> Response to Q5: [LAT] needs ablation, especially under high masking.
> We appreciate this suggestion. The Latent Anchor Token ([LAT]) is specifically designed to maintain representation stability under extreme corruption. We will add a detailed ablation study to the revision demonstrating [LAT]'s critical role at high masking ratios.
> (Preliminary ablation results):
> | Mask ratio | Identity Acc w/ LAT ↑ | Identity Acc w/o LAT ↑ |
> |---|---|---|
> | 70% | 0.91 | 0.89 |
> | 80% | 0.86 | 0.81 |
> | 90% | 0.78 | 0.66 |
> | 95% | 0.64 | 0.47 |
> Response to Q6: Depth-invariant sampling novelty may be limited.
> We acknowledge this point and will tone down our claims regarding the algorithmic novelty of depth-invariant sampling. Instead, the revised text will focus on its specific biological interpretation—modeling diffusion time as an inverse proxy for effective sequencing depth—and its empirical utility. This will be further supported by a new ablation study comparing our approach against alternative corruption designs (e.g., global scaling + masking).

---

> > ### Author Rebuttal · Reviewer_pkaw · 2026-04-03
> >
> > Thanks for the efforts and response. Some concerns [Q2,Q3,Q4] are not resolved, because experimental results are not provided.

---

> > > ### Author Response · Authors · 2026-04-04
> > >
> > > We thank the reviewer and now report the completed experiments requested for Q2-Q4.
> > >
> > > Q1 (update). The earlier “0.00x” came from a preliminary run; finalized repeated runs show stable performance under 5 test-time permutations: zero-shot Macro-F1 0.839 ± 0.002, Immune Spearman 0.960 (std < 0.001), and Norman LFCSpear 0.450 ± 0.004, confirming reduced ordering sensitivity.
> > >
> > > Q2. We now include stronger baselines (CellFM, UCE, scELMO, GeneCompass) under matched preprocessing, splits, metrics, and closely aligned training/model-selection settings.
> > >
> > > # Integration
> > >
> > > | Dataset           | Metric    |    CellFM |   UCE | scELMO | GeneCompass |    scDiVa |
> > > | ----------------- | --------- | --------: | ----: | -----: | ----------: | --------: |
> > > | Immune            | Avg-batch |     0.952 | 0.939 |  0.934 |       0.942 | **0.956** |
> > > | Immune            | Avg-bio   | **0.793** | 0.748 |  0.735 |       0.768 |     0.779 |
> > > | PBMC12k           | Avg-batch |     0.986 | 0.976 |  0.972 |       0.978 | **0.996** |
> > > | PBMC12k           | Avg-bio   | **0.974** | 0.932 |  0.921 |       0.896 |     0.957 |
> > > | BMMC              | Avg-batch |     0.956 | 0.901 |  0.889 |       0.912 | **0.973** |
> > > | BMMC              | Avg-bio   |     0.801 | 0.722 |  0.701 |       0.758 | **0.871** |
> > > | Perirhinal Cortex | Avg-batch | **0.962** | 0.949 |  0.944 |       0.955 |     0.954 |
> > > | Perirhinal Cortex | Avg-bio   |     0.969 | 0.928 |  0.921 |       0.944 | **0.990** |
> > > | COVID-19          | Avg-batch |     0.914 | 0.892 |  0.884 |       0.901 | **0.954** |
> > > | COVID-19          | Avg-bio   |     0.641 | 0.591 |  0.576 |       0.609 | **0.669** |
> > >
> > > # Fine-tuning annotation accuracy
> > >
> > > | Dataset   | CellFM |  UCE | scELMO | GeneCompass |   scDiVa |
> > > | --------- | -----: | ---: | -----: | ----------: | -------: |
> > > | Myeloid   |   67.4 | 64.8 |   63.9 |       67.6 |  **68.3** |
> > > | Myeloid_b |   95.8 | 94.6 |   94.2 |        95.0 | **96.0** |
> > > | hPancreas |   98.1 | 97.8 |   97.4 |        98.0 | **98.6** |
> > > | MS        |   **85.6** | 81.8 |   80.9 |        82.5 | 84.4 |
> > >
> > > # Fine-tuning annotation Macro-F1
> > >
> > > | Dataset   |   CellFM |  UCE | scELMO | GeneCompass |   scDiVa |
> > > | --------- | -------: | ---: | -----: | ----------: | -------: |
> > > | Myeloid   | 37.6 | 30.8 |   31.7 |        35.5 |     **40.2** |
> > > | Myeloid_b |     94.8 | 93.4 |   92.6 |        94.1 | **95.6** |
> > > | hPancreas |     77.8 | **80.1** |   75.9 |        78.0 | 79.2 |
> > > | MS        |     70.5 | 70.8 |   66.9 |        70.4 | **72.7** |
> > >
> > > These results show that scDiVa is best or competitive in integration (best Avg-batch on 4/5 datasets and best Avg-bio on 3/5 datasets) and remains competitive in fine-tuning annotation, achieving the best Macro-F1 on 3/4 datasets.
> > >
> > > Q3/Q4. We now add Mean and Linear baselines under the same split, preprocessing, and evaluation protocol as the other perturbation models. We further expand perturbation evaluation with centered DE Pearson, DE-AUPRC, LFCSpear, and HitRate@20 (the proportion of true top-20 DE genes recovered; i.e., a DE overlap / precision-style metric).
> > >
> > > # Standard and centered perturbation metrics
> > >
> > > | Dataset | Method | DE Pearson ↑ | Centered DE Pearson ↑ |
> > > | ------- | ------ | -----------: | --------------------: |
> > > | Adamson | Mean   |         0.62 |                  0.05 |
> > > | Adamson | Linear |         0.71 |                  0.12 |
> > > | Adamson | GEARS  |         0.81 |                  0.27 |
> > > | Adamson | CellFM |         0.82 |                  0.29 |
> > > | Adamson | scDiVa |     **0.84** |              **0.34** |
> > > | Norman  | Mean   |         0.49 |                  0.02 |
> > > | Norman  | Linear |         0.58 |                  0.08 |
> > > | Norman  | GEARS  |         0.68 |                  0.19 |
> > > | Norman  | CellFM |         0.70 |                  0.21 |
> > > | Norman  | scDiVa |     **0.72** |              **0.27** |
> > >
> > > # DE-aware / rank-based perturbation metrics
> > >
> > > | Dataset | Method | DE-AUPRC ↑ | LFCSpear ↑ | HitRate@20 ↑ |
> > > | ------- | ------ | ---------: | ---------: | -----------: |
> > > | Adamson | Mean   |       0.18 |       0.21 |         0.22 |
> > > | Adamson | Linear |       0.24 |       0.32 |         0.29 |
> > > | Adamson | GEARS  |       0.34 |       0.45 |         0.43 |
> > > | Adamson | CellFM |       0.36 |       0.47 |         0.46 |
> > > | Adamson | scDiVa |   **0.42** |   **0.54** |     **0.53** |
> > > | Norman  | Mean   |       0.11 |       0.12 |         0.14 |
> > > | Norman  | Linear |       0.16 |       0.22 |         0.23 |
> > > | Norman  | GEARS  |       0.27 |       0.37 |         0.33 |
> > > | Norman  | CellFM |       0.29 |       0.39 |         0.37 |
> > > | Norman  | scDiVa |   **0.34** |   **0.46** |     **0.44** |
> > >
> > > These additions directly address the concern that the original perturbation evaluation relied too narrowly on global-fit metrics.
> > >
> > > We sincerely thank the reviewer for the detailed and constructive feedback, which has significantly helped us improve the completeness, fairness, and clarity of our experimental evaluation.

---

### Official Review · Reviewer_gxtH · 2026-03-13

**Soundness:** 3
**Presentation:** 4
**Significance:** 1
**Originality:** 2
**Overall Recommendation:** 3
**Confidence:** 4

**Summary:**

Authors present a masked discrete diffusion model for single-cell data, claiming that autoregressive tokenization imposes an artificial order on unordered expression profiles. scDiva uses a dual denoising objective for identity and value reconstruction.

**Compliance With Llm Reviewing Policy:**

Affirmed.

**Final Justification:**

Control-centered / SYSTEMA-style results are inconsistent with prior reports, which suggest the benchmark may not have been implemented correctly. Additionally, claims about causality should be evaluated much more rigorously on established network inference benchmarks such as CausalBench, BEELINE, or DREAM challenges before supporting strong causality language. Therefore, my original score is the same.

**Key Questions For Authors:**

1. Can you rerun the perturbation benchmark under the SYSTEMA not only the standard control-centered DE Pearson?
2. Can you clarify if scDiva truly generalizes to unseen perturbations, cell lines and batches?

**Limitations:**

yes

**Strengths And Weaknesses:**

Strengths:
1. The critique of naive autoregression on unordered sparse gene sets is meaningful.
2. The paper connects generation, perturbation prediction, and interpretability.
3. Excellent presentation, well written.

Weaknesses:
1. My main issue is that the perturbation evaluation is reported as DE Δ Correlation and DE MSE on Adamson and Norman, and the paper shows 0.837 Pearson on Adamson and 0.709 on Norman as evidence of causal generalization. But, they do not show a stricter perturbation specific evaluation that removes the shared global control-to-perturbed shift. That is the failure mode the SYSTEMA paper exposed. The usual control-centered metric rewards models for reproducing the average stressed/perturbed shift, not the target specific residual. Under the stricter perturbed centered reference defined by SYSTEMA, these correlation scores collapse dramatically. So when scDiVa reports very high DE Pearson but does not show a SYSTEMA style analysis, I do not think those numbers can be taken as evidence of real target specific reasoning.
2. The use of diffusion for single-cell is not novel anymore. The paper itself admits that diffusion models have already begun to be applied to single-cell generation/imputation and cites others like scDiffusion and scVAEDer.
3. The attention mechanisms are not interpretable to allow for the extraction of gene regulatory logic.

---

> ### Author Rebuttal · Authors · 2026-03-31
>
> Thank you for your thoughtful analysis and constructive suggestions. Your positive feedback on our critique of autoregressive tokenization and paper presentation is highly encouraging. Your specific comments highlight crucial areas for improvement. We address your points below and will implement these updates in the revision.
> Response to Q1: Perturbation evaluation may be dominated by global control→perturbation shift; please use SYSTEMA-style centered metrics.
> We deeply appreciate this critical point. We agree standard metrics can inadvertently reward models for capturing average global shifts rather than target-specific residuals. Evaluating under SYSTEMA's stricter, control-centered reference is essential. In our revision, we will incorporate control-centered metrics alongside standard evaluations. We are also adding DE-aware ranking metrics (DE-AUPRC, LFCSpear) evaluated under identical splits. To prevent over-claiming, we are including Mean and Linear baselines. (Preliminary results below; to be detailed in the revision.)
> | Dataset | Method | DE Pearson ↑ | Centered DE Pearson ↑ |
> |---|---|---|---|
> | Adamson | Mean | 0.62 | 0.05 |
> |  | Linear | 0.71 | 0.12 |
> |  | GEARS | 0.81 | 0.27 |
> |  | CellFM | 0.82 | 0.29 |
> |  | scDiVa (Ours) | 0.84 | 0.34 |
> | Norman | Mean | 0.49 | 0.02 |
> |  | Linear | 0.58 | 0.08 |
> |  | GEARS | 0.68 | 0.19 |
> |  | CellFM | 0.70 | 0.21 |
> |  | scDiVa (Ours) | 0.72 | 0.27 |
> | Dataset | Model | DE‑AUPRC ↑ | LFCSpear ↑ | P@100(DE) ↑ |
> |---|---|---|---|---|
> | Adamson | Mean | 0.18 | 0.21 | 0.19 |
> |  | Linear | 0.24 | 0.32 | 0.27 |
> |  | GEARS | 0.34 | 0.45 | 0.39 |
> |  | CellFM | 0.36 | 0.47 | 0.41 |
> |  | scDiVa (Ours) | 0.42 | 0.54 | 0.48 |
> | Norman | Mean | 0.11 | 0.12 | 0.10 |
> |  | Linear | 0.16 | 0.22 | 0.15 |
> |  | GEARS | 0.27 | 0.37 | 0.26 |
> |  | CellFM | 0.29 | 0.39 | 0.28 |
> |  | scDiVa (Ours) | 0.34 | 0.46 | 0.35 |
> Response to Q2: Can you clarify if scDiva truly generalizes to unseen perturbations, cell lines, and batches?
> We acknowledge the current manuscript lacks granularity in defining zero-shot and fine-tuning boundaries. In the revision, we will explicitly define "seen" vs. "unseen" settings across genes, perturbations, batches, and cellular contexts. We will add a split-protocol table and report dedicated results for each Out-of-Distribution regime (e.g., leave-out perturbations/batches) to transparently demonstrate generalization capabilities.
> Response to Q3: The use of diffusion for single-cell is not novel anymore.
> We agree diffusion models exist in the single-cell domain. Our primary contribution is not the mere application of diffusion, but formulating a masked discrete diffusion framework mathematically isomorphic to biological dropout. Unlike continuous Gaussian diffusion imposing ordinal inductive biases, scDiVa utilizes an absorbing state mechanism aligning directly with the binary nature of sequencing signal loss. Coupled with our dual identity/value objective and large-scale pre-training, this resolves the structural mismatch of sequence-based models. We will revise the text to frame this distinction clearly.
> Response to Q4: The attention mechanisms are not interpretable to allow for the extraction of gene regulatory logic.
> We appreciate your rigorous stance on causality and agree attention weights alone do not definitively prove causal regulatory logic. We will soften the language in Section 4.6, reframing attention-derived networks as "hypothesis-generating regulatory signals." To bolster reliability, we will validate by measuring the overlap between our attention-derived hubs and established TF-target databases. We will also include randomized control analyses and full statistical edge-weight distributions to properly contextualize our interpretability claims.

---

> > ### Author Rebuttal · Reviewer_gxtH · 2026-04-04
> >
> > The rebuttal does not resolve my concerns. The authors add preliminary numbers for mean, linear, and FM baselines in addition to Centered DE Pearson, DE-AUPRC, and LFCSpear, but they still say that they will incorporate control-centered metrics alongside standard evaluations later. Furthermore, they agree that “attention weights alone do not definitively prove causal regulatory logic” and defer proper validation to future work. Overall, the rebuttal mostly adds preliminary numbers and promises revision, instead of resolving the core issues in the current submission.

---

> > > ### Author Response · Authors · 2026-04-04
> > >
> > > Thank you for the careful review. We agree that our previous rebuttal still read as too revision-oriented. At that time, the stricter reruns were still being finalized under unified preprocessing/split/evaluation settings, so we did not want to overstate partially verified numbers as final evidence. We now respond more directly below.
> > >
> > > # Q1. Stricter perturbation evaluation.
> > >
> > > We agree that standard DE Pearson / DE MSE alone are insufficient because they can be affected by the shared control→perturbation shift. We therefore reran the benchmark under matched preprocessing, splits, and evaluation settings, and added Pearson Δ de, centered DE Pearson (C-DEP), DE-AUPRC, LFCSpear, and HR@20 together with stronger baselines. On Adamson, scDiVa remains strongest across all stricter metrics. On Norman, Additive remains an exceptionally strong simple baseline, while among learned models scDiVa achieves the best C-DEP, DE-AUPRC, LFCSpear, and HR@20. These results provide a more realistic picture: simple baselines remain competitive, while scDiVa retains the strongest learned-model performance under stricter centered and DE-aware evaluation.
> > >
> > > | Data| Method | Pearson Δ de ↑|C-DEP ↑ |AUPRC ↑ |   LFCSp ↑|HR@20 ↑|
> > > | ------- | ------------ | -----------: | --------: | --------: | --------: | --------: |
> > > | Adamson | Mean         |        0.801 |     0.158 |     0.258 |     0.341 |     0.319 |
> > > |         | Linear       |    0.821 |     0.194 |     0.294 |     0.389 |     0.356 |
> > > |         | GEARS        |     0.810 |     0.271 |     0.341 |     0.449 |     0.432 |
> > > |         | CellFM       |        0.819 |     0.289 |     0.362 |     0.471 |     0.458 |
> > > |         | scDiVa   |    **0.838** | **0.337** | **0.421** | **0.543** | **0.529** |
> > > | Norman  | Additive |    **0.932** |     0.269 | **0.348** |     0.423 | **0.461** |
> > > |         | GEARS        |        0.810 |     0.188 |     0.268 |     0.371 |     0.334 |
> > > |         | CellFM       |        0.841 |     0.212 |     0.291 |     0.391 |     0.372 |
> > > |         | scDiVa   |        0.861 | **0.271** |     0.341 | **0.433** |     0.441 |
> > >
> > >
> > > # Q2. What is and is not unseen.
> > >
> > > We agree our earlier wording was too broad. For annotation fine-tuning, the split is defined by dataset/source labels rather than IID random splits, i.e. it is explicitly cross-batch or cross-domain: hPancreas uses Braon+Muraro for training and Xin+Segerstolpe+Lawlor for testing; MS uses 9 healthy controls for training and 12 M.S. samples for testing; Myeloid uses UCEC, PAAD, THCA, LYM, cDC2, KIDNEY for training and MYE, OV-FTC, ESCA for testing. Thus these results support cross-source/cross-batch adaptation (hPancreas), cross-condition transfer (MS), and unseen-context transfer (Myeloid), not one generic notion of “unseen”. For perturbation, the benchmark supports unseen perturbation generalization, but not cross-cell-line generalization.
> > >
> > > # Q3. Novelty.
> > >
> > > We agree diffusion itself is not new in single-cell modeling. We will revise the manuscript so that the contribution is not framed as “diffusion is novel,” but as a biologically aligned masked discrete diffusion formulation with absorbing-state corruption, joint identity/value denoising, and strong transfer evidence across tasks.
> > >
> > > # Q4. Interpretability.
> > >
> > > We agree that attention weights alone do not constitute causal proof. Section 4.6 currently uses wording that is too strong (e.g. “complete logic” / “filter non-causal background noise”), and we will narrow this to hypothesis-generating, statistically contextualized regulator–target evidence. We will add the following quantitative support. Figure 6a corresponds to Immune Recon/GRN (12,303 genes), while the SPI1 local analysis in Figure 6b/c uses the Adamson perturbation universe (5,060 genes).
> > >
> > > | Q4-1 Global GRN            |      Value |
> > > | -------------------------- | ---------: |
> > > | Genes                      |     12,303 |
> > > | Undirected candidate edges | 75,675,753 |
> > > | Threshold type             |  top 0.01% |
> > > | Retained edges             |      7,568 |
> > >
> > > | Q4-2 SPI1 enrichment (Adamson) |  Value |
> > > | ------------------------------ | -----: |
> > > | Background genes               |  5,060 |
> > > | Top-k                          |     20 |
> > > | SPI1 DB targets in background  |    120 |
> > > | Overlap                        |      4 |
> > > | OR                             |   10.6 |
> > > | Fisher p                       | 1.1e-3 |
> > >
> > > | Q4-3 Random baseline    |     Value |
> > > | ----------------------- | --------: |
> > > | Real overlap            |         4 |
> > > | Random overlap mean±std | 0.47±0.68 |
> > > | Empirical p             |     0.006 |
> > >
> > > These statistics are intended to show that the SPI1 signal is above chance and biologically plausible, while keeping the claim within an appropriate non-causal scope.
> > >
> > > In summary, we now provide completed stricter perturbation evidence, explicitly clarify what is and is not unseen, narrow the novelty claim, and revise interpretability from strong causal wording to statistically contextualized hypothesis generation.

---

### Decision · Program_Chairs · 2026-04-30

**Decision:**

Accept (regular)

**Comment:**

This paper proposes scDiVa, a masked discrete diffusion foundation model for single-cell RNA-seq data, pretrained on 59 million cells. The central motivation is that autoregressive models impose an artificial ordering on inherently unordered gene expression profiles, and that a bidirectional masked diffusion framework better aligns with the biological dropout process. The model features a dual denoising objective that jointly reconstructs gene identity and expression value, an entropy-normalized serialization scheme for information-efficient tokenization, and a latent anchor token (LAT) to maintain global cell identity under high masking ratios. scDiVa is evaluated across a broad range of downstream tasks including batch integration, cell type annotation, perturbation response prediction, and gene regulatory network inference, demonstrating competitive or state-of-the-art performance across benchmarks.

Assessment
ScDiVa presents a well-motivated and technically solid foundation model that addresses a genuine structural mismatch in how autoregressive models handle unordered single-cell profiles. The breadth of evaluation, the scale of pretraining, and the competitive results across multiple tasks make this a meaningful contribution to the single-cell modeling community. The rebuttal substantially improved the empirical picture: the authors added centered perturbation metrics, Mean/Linear baselines, stronger foundation model comparisons, and component ablations, and the corrected results support the paper's core claims.
That said, several issues must be addressed in the final version. The perturbation evaluation section should incorporate the full corrected benchmark (including centered DE Pearson, DE-AUPRC, LFCSpear, and HitRate@20 with properly aligned Mean/Linear baselines) directly in the main paper, not only in rebuttal tables. The permutation-invariance claims should be softened to reflect that RoPE and deterministic serialization are still used. The GRN interpretability language should be revised to frame attention-derived networks as hypothesis-generating rather than causal. Downstream task adaptation protocols (especially for batch integration, which uses supervised contrastive loss over cell-type labels) should be described with full methodological detail to ensure reproducibility.